Prepared for submission to JHEP

# Regge Limit of One-Loop String Amplitudes

**Pinaki Banerjee**[1,3]**, Lorenz Eberhardt**[2]**, Sebastian Mizera**[3]

[1]*ICTP South American Institute for Fundamental Research, IFT-UNESP, São Paulo, SP Brazil 01440-070*

[2]*Institute for Theoretical Physics, University of Amsterdam, Amsterdam, 1098XH, NL*

[3]*Institute for Advanced Study, Einstein Drive, Princeton, NJ 08540, USA*

*E-mail:* pinaki@ictp-saifr.org, l.eberhardt@uva.nl, smizera@ias.edu

ABSTRACT: We study the high-energy limit of $2 \to 2$ one-loop string amplitudes at fixed momentum transfer. For the closed string, the high-energy behaviour of the amplitudes can be determined from Regge theory just like in field theory, as was first discussed by Amati, Ciafaloni and Veneziano. However, field theory intuition partially breaks down for the open-string amplitude, where amplitudes can exhibit surprising asymptotics in the high-energy limit depending on the topology of the diagram. We call this phenomenon *Regge attenuation*. We extract Regge limits by a combination of unitarity cuts and saddle-point analysis. We show that the leading contribution of the planar open-string amplitude is sufficiently simple that we can extract it at *any* loop order. This allows us to resum the genus expansion in a certain limit and demonstrate that the leading Regge trajectory remains linear in that limit.

## 1  Introduction

Scattering amplitudes simplify drastically in the Regge limit. In this limit, we consider $2 \to 2$ scattering at very large energies $s > 0$ while keeping the momentum transfer $t < 0$ fixed. General arguments involving analyticity in the spin of the partial wave decomposition allow one to show under reasonable assumptions that any amplitude behaves in that limit as [1, 2]

$$A(s,t) \sim \beta(t)\, s^{\alpha(t)} \tag{1.1}$$

for some functions $\alpha(t)$ and $\beta(t)$, up to subleading corrections. The exponent $\alpha(t)$ contains a lot of physics. For example, for tree-level string amplitudes, we get *linear* Regge trajectories,

$$\alpha(t) = J + \alpha' t \ . \tag{1.2}$$

Here $\alpha'$ is the Regge slope (and gives $\alpha'$ its name), while $J$ is the Regge intercept. It coincides with the highest spin of a massless particle in the theory, i.e. $J = 1$ for open strings and $J = 2$ for closed strings.

In this paper, we will highlight the difference between the point-particle and stringy Regge analysis by studying the behavior of one-loop string amplitudes. We study the limit of large $\alpha'|s|$ while keeping $\alpha'|t|$ finite. Of course, this is a slightly different limit since we are working at weak string coupling and take the Regge limit at a fixed loop order, instead of keeping the string coupling $g_{\mathrm{s}}$ finite. Nevertheless, we view this as a step towards achieving the more ambitious goal of understanding the Regge limit at finite coupling.

It is also interesting that string amplitudes simplify drastically in this limit. Even at one-loop level, they become simple combinations of Gamma functions, see e.g. (4.6) for the closed-string result. It seems possible that there is a good expansion of the amplitude in the Regge limit in $\frac{1}{s}$, with the strict asymptotic expression being the leading term. Such a high-energy expansion seems much simpler to us than the more conventional low-energy or $\alpha'$ expansion that is useful when comparing to effective field theory and which has been studied extensively in the literature (see e.g. [3, 4] for the case of closed string one-loop amplitudes). The low-energy expansion leads

| type | scaling | contributing cuts |
|---|---|---|
| **closed** (4.6, 4.28) | $\sim i\dfrac{s^{3+\alpha'_c t/2}}{\log^4(s)}$ | |
| **open (1234)** (6.8) | $\sim i\, s^{2+\alpha' t}$ | |
| **open (1342)** (5.11, 5.34) | $\sim i\dfrac{s^{1+\alpha' t/2}}{\log^4(s)}$ | |
| **open (1423)** (6.10) | $\sim i\, s^{2+\alpha' t}$ | |
| **open (12)(34)** (5.11, 5.22) | $\sim i\dfrac{s^{1+\alpha' t/2}}{\log^4(s)}$ | |
| **open (13)(24)** (5.11, 5.26) | $\sim i\dfrac{s^{1+\alpha' t/2}}{\log^4(s)}$ | |
| **open (14)(23)** (5.43) | $\sim s^{2+\alpha' t/2}$ | |

**Figure 1:** Summary of the Regge behavior of one-loop string amplitudes in the $s$-channel for $t < 0$ and for real $s \gg 0$. For open string amplitudes we indicate the planar ordering under consideration. Behavior indicates the leading Regge scaling without prefactors ($i$ indicates that the leading behavior is purely imaginary), after including the polarization tensors $t_8$ and $t_8\tilde{t}_8$. On the right, we display all unitarity cuts contributing to a given topology, where shadings indicate which contributions are dominant in the Regge limit. Lack of shading means that the real part dominates instead. The two planar orderings (1234) and (1324) are a bit special and do not fit into this terminology. We refer to Sections 4, 5 and 6 for details.

to rich number-theoretic objects known as modular graph forms [5, 6]. See [7] for a summary of the state-of-the-art computations.

We will obtain our results from two different techniques that are useful to analyze the high-energy limit. They are complementary and both only reveal partial information. We however find a coherent picture and are hence confident about the validity of our findings. They are summarized in Figure 1.

The first technique was pioneered by Gross and Mende [8, 9] for the evaluation of the fixed angle high-energy limit of the string amplitude ($s, t \to \infty$ with $s/t$

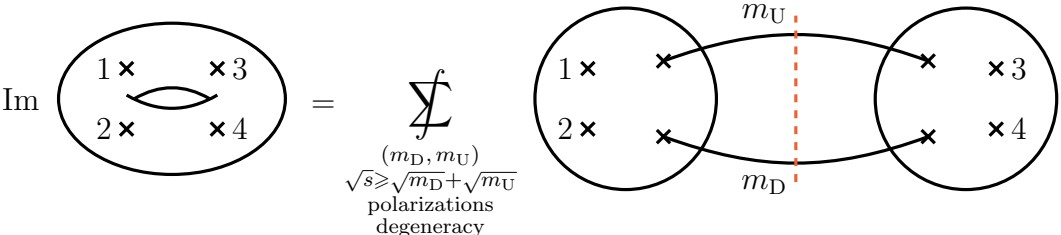

**Figure 2:** Unitarity cut of the genus-one closed string amplitude.

fixed). One notices that for large Mandelstam variables, the moduli space integral is dominated by certain special geometries and the integral can be evaluated via saddle-point approximation. Such a saddle-point approximation is technically very hard to carry out explicitly. For the general case, there are many unresolved problems with this approach, such as (i) there is no general classification of the saddles that contribute to the integral, (ii) the moduli space integral does not actually converge and first has to be deformed into a steepest descent contour, (iii) there can be flat directions as well as intersecting saddles and other phenomena that make it challenging to directly apply saddle point approximation.

The structure of the saddle-point approximation is however simpler in the case of the Regge limit that we study in this paper. It was demonstrated by Sundborg in [10] that for purely *imaginary* choices of the Mandelstam variable $s$ where the moduli space integral is convergent, the saddle-point evaluation can be explicitly carried out for the closed string one-loop amplitude. One can then attempt to analytically continue the result back to real values of $s$, but this analytic continuation is in general not unique due to Stokes phenomena and hence only gives partial results.

The second technique computes the imaginary part of the amplitude from unitarity cuts. They can be directly derived from the worldsheet integral representation and schematically write the imaginary part of the amplitude as a square of the tree-level amplitude, as in Figure 2. One can make this formula very explicit for string amplitudes and compute the Regge limit of every possible pair of masses-squared $(m_\mathrm{D}, m_\mathrm{U})$ crossing the cut. Not surprisingly, every term behaves like a field theory amplitude in the Regge limit. We then attempt to resum the individual contributions across all possible $(m_\mathrm{D}, m_\mathrm{U})$. It turns out that this is possible in some cases, but not all.

For the closed string, this procedure works and is consistent with the result obtained from saddle-point approximation. For the closed string, this method is also equivalent to the Regge methods [11] that were used by Amati, Ciafaloni and Veneziano to obtain the same results [12, 13]. For the open string, this procedure only works some of the times, depending on the topology of the diagram.

We call this phenomenon *Regge attenuation*. It sharply demonstrates the difference of field theory and string amplitudes. For example, we see explicitly that the non-

planar annulus diagram in an appropriate channel is dominated by the exchange of tree-level gravitons in the dual closed-string channel, which leads to an unexpected Regge behaviour from the field theory point of view. There are further surprises for the open string. We find that the planar one-loop amplitude grows like $i\,s^{2+\alpha't}$ at one-loop, instead of $i\,s^{1+\alpha't}\log(s)$, which would be the expected behaviour from field theory. This goes against the common lore in the subject and we already noticed this behaviour before in [14, 15]. Part of the motivation for the present paper is to understand this tension with the field theory intuition.

The resolution turns out to be simple, but inherently stringy in nature. The string theory tree-level amplitude has an infinite number of poles coming from the massive resonances. The tree-level contribution takes the following form in the Regge limit,

$$A_{\text{tree}}(s,t) \sim -\frac{\sin(\pi\alpha'(s+t))}{\sin(\pi\alpha's)}\,\Gamma(-t)\,(\alpha's)^{1+\alpha't}\,. \tag{1.3}$$

The presence of the additional trigonometric factor in this formula with respect to (1.1) has the physical interpretation of an infinite number of resonances at integer values of $\alpha's$. It has an important consequence. At one-loop level, $\alpha'$ gets renormalized to $\alpha'+\delta\alpha'$. Among other features, this effect produces a term in the Regge limit that is proportional to the $\alpha'$ derivative of this expression. The $\alpha'$ derivative of the last term gives a contribution to the one-loop amplitude that has a $\log(s)$ enhancement. However, the $\alpha'$ derivative can also hit the trigonometric function, which gives a contribution that has a whole power of $s$ enhancement. It is precisely this contribution that becomes leading for the planar amplitude at one loop.

This contribution turns out to be simple enough that we can isolate it at *any* loop order in the loop expansion of the planar amplitude. We are thus able to resum the loop expansion in the double-scaled limit

$$s \to \infty\,, \qquad g_{\text{s}}^2 s = \text{const.} \tag{1.4}$$

and find that the amplitude behaves like the tree-level amplitude, but with a leading linear Regge trajectory that is slightly rotated into the imaginary plane, see eq. (6.20).

This paper is organized as follows. In Section 2, we recall the basics of Regge theory and their implications for string amplitudes. We then state formulas for the imaginary part of the one-loop amplitude in Section 3 for both the open and closed string. They haven't appeared in this form before in the literature and we included a derivation in the closed string case in Appendix A, while the derivation of the open string rules are discussed in Appendix B. The Regge behaviour of the closed string amplitude is discussed in Section 4. This section does not contain new results, but reviews and streamlines the derivation of [10, 13]. We finally discuss the main case of interest in Section 5, where we show that saddle point approximation and Regge

techniques determine the one-loop behaviour of a variety of string diagrams. The planar annulus cannot be analyzed in this way and we discuss it in detail in Section 6. We end with a discussion and future directions in Section 7.

## 2  When is the Regge limit dominated by cuts?

Analyticity of scattering amplitudes puts some interesting constraints on their asymptotic behavior. In this section, we review the necessary background to understand this connection. In particular, we will focus on the question whether the Regge limit is dominated by real or imaginary contributions, see [16, 17] for previous discussions in the context of perturbative QCD. The answer to this question will help us understand if computing the Regge behavior $\text{Im}\,\mathcal{A}(s,t)$ using unitarity cuts is sufficient to determine the Regge behavior of $\mathcal{A}(s,t)$.

### 2.1  Naive argument

We first focus on the $s$-channel kinematics with $s > -t > 0$. The main idea is that the high-energy behavior in the $s$-channel, $s \gg -t > 0$, will be dominated by an exchange of states in the $t$-channel. For this reason, we start by performing the $t$-channel partial-wave expansion of $\mathcal{A}(s,t)$,

$$\mathcal{A}(s,t) = \sum_{\ell=0}^{\infty} (2\ell + 1)\, a_\ell(t)\, G_\ell(z_t)\,, \tag{2.1}$$

where $z_t$ is the cosine of the scattering angle $\theta_t$ equal to $z_t = \cos\theta_t = 1 + 2s/t$ for massless external states. In the above equation, $a_\ell(t)$ are the $t$-channel partial-wave amplitudes and $G_\ell = C_\ell^{((D-3)/2)}$ are the Gegenbauer polynomials, which are the counterparts of Legendre polynomials in D space-time dimensions. The angular momentum $\ell$ is often called partial-wave *spin*, but it should not be confused with the fundamental spin of particles or strings: it is the angular momentum of the whole state exchanged in the $t$-channel, which could be a composite multi-particle state.

If only a state with angular momentum $\ell = J$ is exchanged in the $t$-channel, the amplitude would go as $\mathcal{A}(s,t) \sim (2J + 1)a_J(t)\, G_J(1 + 2s/t)$. In the Regge limit, $z_t \to -\infty$. We can use the known behavior of Gegenbauer polynomials, which is $G_\ell(z_t) \sim (-z_t)^\ell$ up to prefactors. This immediately tells us that a contribution from the $J$ partial wave contributes to the asymptotics of the amplitude as

$$\mathcal{A}(s,t) \sim a_J(t)\left(\frac{s}{-t}\right)^J\,. \tag{2.2}$$

In other words, exchanges of a spin-$J$ state in the $t$-channel make the amplitude go as $s^J$ in the Regge limit.

The above argument, while slick, is not actually completely correct. It is because the $t$-channel partial-wave expansion does not converge in the $s$-channel kinematics.

In fact, one can show that (2.1) converges only in the interval $-1 < z_t < 1$ (the limiting case of the so-called Lehmann ellipse), while the $s$-channel has $z_t < -1$. A more careful treatment requires the theory of complex angular momenta, or Regge theory, which we turn to next.

## 2.2 Basics of Regge theory

The basic idea is to treat the partial-wave spin $\ell$ as a complex variable, similar to the way we complexify the kinematic invariants $s$ and $t$. At first glance, such extension to $\ell \in \mathbb{C}$ does not seem unique. This is because we could have shifted $a_\ell \to a_\ell + f(\ell)$ for any function $f(\ell)$ that vanishes when $\ell = 0, 1, 2, \ldots$. However, a powerful result in complex analysis called Carlson's theorem tells us that $f(\ell) = 0$ identically, provided that

$$|a_\ell| < C \, \mathrm{e}^{\pi|\ell|} \tag{2.3}$$

for some constant $C$. In other words, if partial-wave amplitudes are bounded exponentially at large $\ell$, their extension to the complex plane is unique.

The caveat in this discussion is that $a_\ell$ does not actually satisfy the above bound. In fact, a little detour into the Froissart–Gribov formula would show that $a_\ell$ have to have an alternating sign $(-1)^\ell$. This is not consistent with the above bound since $a_\ell \sim \mathrm{e}^{-i\pi\ell}$ would grow as $\sim \mathrm{e}^{\pi \, \mathrm{Im} \, \ell}$ along the imaginary axis and hence barely violate (2.3). In order to get around this problem, we separate the even and odd terms in $\ell$. We define

$$\mathcal{A}^\pm(s,t) = \frac{1}{2} \sum_{\ell=0}^\infty (2\ell + 1) \, a_\ell(t) \left[ G_\ell(z_t) \pm G_\ell(-z_t) \right] . \tag{2.4}$$

Note that this formula applies to external scalars, while in the string theory case we will study external gluon and graviton supermultiplets. However, the discussion is not affected by this detail as we can consider looking at the scalar part of each multiplet, see [18] for further discussion of this point.

The expansion (2.4) requires some care when the kinematics is tuned to a resonance, e.g., when $s$ or $u$ is a non-negative integer in the superstring case. Around those points, (2.4) should be thought of as being approached from the $s$ upper half-plane in the $s$-channel and likewise the lower half-plane in the $u$-channel.

Using the symmetry of the Gegenbauer polynomials, $G_\ell(-z_t) = (-1)^\ell G_\ell(z_t)$, we see that only even spins contribute to $\mathcal{A}^+$ and likewise only odd ones contribute to $\mathcal{A}^-$. The full amplitude is the sum, $\mathcal{A} = \mathcal{A}^+ + \mathcal{A}^-$. Carlson's theorem guarantees that $a_\ell$ uniquely extend to complex values of $\ell$, for even and odd terms separately. Let us call these analytic continuations $a^+(\ell, t)$ and $a^-(\ell, t)$ respectively. Note that likewise $G_\ell(\pm z_t)$ can be extended to the complex $\ell$-plane, as given by their definitions in terms of the hypergeometric function:

$$G_\ell(z_t) = \frac{(\mathrm{D} - 3)_\ell}{\Gamma(\ell + 1)} \, {}_2F_1\left(-\ell, \mathrm{D} - 3 + \ell; \tfrac{\mathrm{D} - 2}{2}; \tfrac{1 - z_t}{2}\right) , \tag{2.5}$$

where $(x)_n$ is the Pochhammer symbol. In this representation, $G_\ell(z_t)$ has a branch cut on the negative real axis, going from $z_t = -1$ to $-\infty$, which should be approached from the lower half-plane. Likewise, $G_\ell(-z_t)$ has a branch cut extending between $z_t = 1$ and $\infty$ and the physical approach is from the upper half-plane. In fact, it will be useful to note that asymptotically $G_\ell(z_t) \sim e^{-i\pi\ell} G_\ell(-z_t)$ for complex values of $\ell$, which means we can write

$$G_\ell(z_t) \pm G_\ell(-z_t) \sim [e^{-i\pi\ell} \pm 1] G_\ell(-z_t) \tag{2.6}$$

in the lower half-plane of $z_t$. This relation will become quite central in a moment.

Notice that we can assign a different meaning to the quantities $\mathcal{A}^\pm$. This is because replacing $z_t \to -z_t = 1 + 2u/t$ is the same as replacing $s \to u$. Hence, the second term in the square brackets in (2.4) is the partial-wave expansion of $\mathcal{A}(u,t)$. Therefore, the even and odd amplitudes $\mathcal{A}^\pm$ are simply symmetrized and anti-symmetrized versions of the amplitude under $s \leftrightarrow u$:

$$\mathcal{A}^\pm(s,t) = \frac{1}{2}\left[\mathcal{A}(s,t) \pm \mathcal{A}(u,t)\right] . \tag{2.7}$$

These quantities appear naturally in Regge theory through a requirement of unique continuation in complex angular momentum.

At the core of Regge theory is the Sommerfeld–Watson transform, which is an imaginative rewriting of (2.4) as a contour integral in the complex $\ell$-plane:

$$\mathcal{A}^\pm(s,t) = -\frac{i}{4}\int_\mathcal{H} d\ell\, \frac{(2\ell+1)\, a^\pm(\ell,t)\left[G_\ell(z_t) \pm G_\ell(-z_t)\right]}{\sin(\pi\ell)} . \tag{2.8}$$

Here, $\mathcal{H}$ is the Hankel contour encircling the positive real axis. The equality to (2.4) is simple to demonstrate by closing up this contour around the poles at $\ell = 0, 1, 2, \ldots$. The reason why writing (2.8) is useful is that by deforming the integration contour, we can improve upon the convergence as a function of $z_t$ and hence ultimately extend it to values $z_t < -1$ required to understand the $s$-channel physics.

More concretely, we are going to deform $\mathcal{H}$ into the contour $-\frac{1}{2} + i\mathbb{R}$ running up parallel to the imaginary axis. One can show that the arcs at infinity do not contribute. However, $a^\pm(\ell,t)$ might have a non-trivial analytic features in the complex plane of $\ell$. For the time being, let us assume it has only $k$ simple poles at some positions $\ell = \hat{\alpha}_i^\pm(t)$ for $i = 1, 2, \ldots, k$. Hence, deforming the contour results in the representation

$$\mathcal{A}^\pm(s,t) = -\frac{i}{4}\int_{-\frac{1}{2}-i\infty}^{-\frac{1}{2}+i\infty} d\ell\, \frac{(2\ell+1)\, a^\pm(\ell,t)\left[G_\ell(z_t) \pm G_\ell(-z_t)\right]}{\sin(\pi\ell)}$$

$$+ \frac{\pi}{2}\sum_{i=1}^{k} \frac{(2\ell+1)\, b_i^\pm(t)\left[G_\ell(z_t) \pm G_\ell(-z_t)\right]}{\sin(\pi\ell)}\Bigg|_{\ell=\hat{\alpha}_i^\pm}, \tag{2.9}$$

where $b_i^\pm(t)$ are the residues of $a^\pm(\ell,t)$ at the poles. We assume they are purely real.

## 2.3 Real vs. imaginary part

At this stage, we can take the $s$-channel Regge limit $s \gg -t > 0$, which is equivalent to $z_t \ll -1$. Using (2.6) and the asymptotics of the Gegenbauer polynomials, we get

$$G_\ell(z_t) \pm G_\ell(-z_t) \sim \frac{\Gamma(\ell + \frac{D-3}{2})}{\Gamma(\frac{D-3}{2})\Gamma(\ell+1)}[e^{-i\pi\ell} \pm 1](-2z_t)^\ell. \tag{2.10}$$

This means that the term dominating the asymptotics is the Regge pole with the largest value of $\mathrm{Re}\,\hat{\alpha}_i^\pm(t)$. Let us call it simply $\alpha_i^\pm(t)$ and the corresponding residue $B^\pm(t)$. The function $\alpha^\pm(t)$ is called the *leading Regge trajectory*. One can show that the first line of (2.9), called the background integral, is subleading. This leaves us with the asymptotics

$$\mathcal{A}^\pm(s,t) \sim \frac{1}{2}B^\pm(t)\,\Gamma[-\alpha^\pm(t)]\,[e^{-i\pi\alpha^\pm(t)} \pm 1]\left(\frac{s}{-t}\right)^{\alpha^\pm(t)}. \tag{2.11}$$

We ignored prefactors that are irrelevant to our discussion. The Gamma function prefactor in (2.11) has an infinite number of poles when $\alpha^\pm(t) = 0, 1, 2, \ldots$. These correspond to an infinite number of bound states exchanged in the $t$-channel. Indeed, the polynomial growth of the amplitude in the Regge limit can be attributed to the whole tower of exchanges.

For our purposes, it would be actually more convenient to work with the quantity [16]

$$e^L = \frac{is}{t}, \tag{2.12}$$

which gives $(\frac{s}{-t})^{\alpha^\pm(t)} = e^{\alpha^\pm(t)L}e^{i\pi\alpha^\pm(t)/2}$. After this change of variables, we get:

$$\mathcal{A}^+(s,t) \sim \quad B^+(t)\,\Gamma[-\alpha^+(t)]\,\cos[\pi\alpha^+(t)/2]\,e^{\alpha^+(t)L}, \tag{2.13a}$$

$$\mathcal{A}^-(s,t) \sim -iB^-(t)\,\Gamma[-\alpha^-(t)]\,\sin[\pi\alpha^-(t)/2]\,e^{\alpha^-(t)L}. \tag{2.13b}$$

Hence, once organized in this way, the prefactor is purely real in the case $\mathcal{A}^+$ and purely imaginary for $\mathcal{A}^-$. Recall that this is the result with $s$ slightly in the upper half-plane.

This observation addresses the question we set out to answer. The significance of this statement is that if $\mathrm{Im}\,\mathcal{A}^\pm$ dominates, we can simply determine its value using unitarity, as will be done in the following section.

So far, we have considered only the cases in which the complex $\ell$-plane contained only simple poles. One can show that presence of other types of singularities modifies the story only mildly. More concretely, higher-order poles or branch cuts add extra logarithmic corrections $\sim \log^\beta(s)\,s^\alpha$, where $\beta$ can be determined from the order of the pole or the discontinuity across the cut. We refer to [1, 2] for details.

## 2.4 Tree-level examples

Let us illustrate how the logic of Section 2.3 works in practice on tree-level examples. The simplest case is that of close-string scattering of graviton states given by the Virasoro–Shapiro amplitude:

$$\mathcal{A}_{\text{VS}}(s,t) = t_8 \tilde{t}_8 \frac{\Gamma(-\frac{1}{2}\alpha'_c s)\,\Gamma(-\frac{1}{2}\alpha'_c t)\,\Gamma(-\frac{1}{2}\alpha'_c u)}{\Gamma(1+\frac{1}{2}\alpha'_c s)\,\Gamma(1+\frac{1}{2}\alpha'_c t)\,\Gamma(1+\frac{1}{2}\alpha'_c u)}\,, \tag{2.14}$$

where $s+t+u=0$. We use $\alpha'_c$ to denote the closed-string version of $\alpha'$. Using the known asymptotics of the Gamma function,

$$\Gamma(x) \sim \sqrt{\frac{2\pi}{x}}\, e^{x[\log(x)-1]} \times \begin{cases} 1 & \text{if } x > 0\,, \\ \dfrac{1}{e^{2\pi i x}-1} & \text{if } x < 0\,, \end{cases} \tag{2.15}$$

we find that in the $s$-channel Regge limit we have

$$\mathcal{A}_{\text{VS}}(s,t) \sim 2^{2-\alpha'_c t}\frac{\Gamma(-\frac{1}{2}\alpha'_c t)}{\Gamma(1+\frac{1}{2}\alpha'_c t)}\frac{\sin(\frac{1}{2}\pi\alpha'_c u)}{\sin(\frac{1}{2}\pi\alpha'_c s)}(\alpha'_c s)^{2+\alpha'_c t} \tag{2.16a}$$

$$\sim -2^{2-\alpha'_c t}\frac{\Gamma(-\frac{1}{2}\alpha'_c t)}{\Gamma(1+\frac{1}{2}\alpha'_c t)} e^{-i\pi\alpha'_c t/2}\,(\alpha'_c s)^{2+\alpha'_c t}\,, \tag{2.16b}$$

where we used $t_8 \tilde{t}_8 \sim (\alpha'_c s)^4$. In the second line, we spelled out the asymptotics of the expression when $s$ is in the upper half-plane for which the ratio of sine becomes a phase. The exponent of $s$ is associated with the exchange of a spin-2 particle: this is the graviton, while the slope justifies the definition of $\alpha'_c$. The same asymptotics holds for $\mathcal{A}_{\text{VS}}(u,t)$ because the Virasoro–Shapiro amplitude is $s \leftrightarrow u$ symmetric.

We can now compare this result with the prediction about the imaginary parts. Because of the above exchange symmetry, we have $\mathcal{A}_{\text{VS}}^{-} = 0$ identically. This comes about as $B_{\text{VS}}^{-} = 0$ in eq. (2.13b). Likewise, we also have $\mathcal{A}_{\text{VS}}^{+} = \mathcal{A}_{\text{VS}}$. In this case, the above asymptotics can be reorganized to read

$$\mathcal{A}_{\text{VS}}^{+}(s,t) \sim 2^{2-\alpha'_c t}\frac{\Gamma(-\frac{1}{2}\alpha'_c t)(-\alpha'_c t)^{2+\alpha'_c t}}{\Gamma(1+\frac{1}{2}\alpha'_c t)} e^{(2+\alpha'_c t)L}\,. \tag{2.17}$$

Up to subleading terms, we can therefore identify $\alpha_{\text{VS}}^{+} = 2 + \alpha'_c t$. Once arranged into the above form, the coefficient of $e^{\alpha_{\text{VS}}^{+} L}$ is purely real.

Let us now consider the more interesting example of the Veneziano amplitude, which computes the planar contribution to scattering of four gluons:

$$\mathcal{A}_{\text{V}}(s,t) = t_8\frac{\Gamma(-\alpha' s)\Gamma(-\alpha' t)}{\Gamma(1+\alpha' u)}\,. \tag{2.18}$$

Repeating similar steps, the Regge asymptotics in the $s$-channel is given by

$$\mathcal{A}_{\text{V}}(s,t) \sim \Gamma(-\alpha' t)\frac{\sin(\pi\alpha' u)}{\sin(\pi\alpha' s)}(\alpha' s)^{1+\alpha' t} \tag{2.19a}$$

$$\sim -\Gamma(-\alpha' t)\, e^{-i\pi\alpha' t}\,(\alpha' s)^{1+\alpha' t} \tag{2.19b}$$

in the upper half-plane. On the other hand, for the other planar contribution $\mathcal{A}_V(u,t)$ we get

$$\mathcal{A}_V(u,t) \sim \Gamma(-\alpha't)\,(\alpha's)^{1+\alpha't}\,. \tag{2.20}$$

In both cases, the intercept of the exponent is associated with the exchange of a spin-1 particle, which is the gluon, and the slope is the $\alpha'$.

Computing the symmetrized amplitude, one finds

$$\mathcal{A}_V^+(s,t) \sim i\,\mathrm{e}^{-i\pi\alpha't/2}\,\sin(\pi\alpha't/2)\,\Gamma(-\alpha't)\,(\alpha's)^{1+\alpha't} \tag{2.21a}$$

$$= -\sin(\pi\alpha't/2)\,\Gamma(-\alpha't)\,(-\alpha't)^{1+\alpha't}\,\mathrm{e}^{(1+\alpha't)L}\,. \tag{2.21b}$$

This identifies $\alpha_V^+ = 1 + \alpha't$. In agreement with (2.13a), the coefficient is purely real.

Finally, let us compute the difference:

$$\mathcal{A}_V^-(s,t) \sim -\mathrm{e}^{-i\pi\alpha't/2}\,\cos(\pi\alpha't/2)\,\Gamma(-\alpha't)\,(\alpha's)^{1+\alpha't} \tag{2.22a}$$

$$= -i\,\cos(\pi\alpha't/2)\,\Gamma(-\alpha't)\,(-\alpha't)^{1+\alpha't}\,\mathrm{e}^{(1+\alpha't)L}\,. \tag{2.22b}$$

Here, we find $\alpha_V^- = 1 + \alpha't$ and the coefficient is purely imaginary, as expected from (2.13b).

## 2.5 One-loop Regge trajectories

Consider a tree-level amplitude with Regge trajectory $\alpha(t)$. Let us determine the Regge behaviour of the imaginary part of the one-loop amplitude that follows from the following unitarity cut:

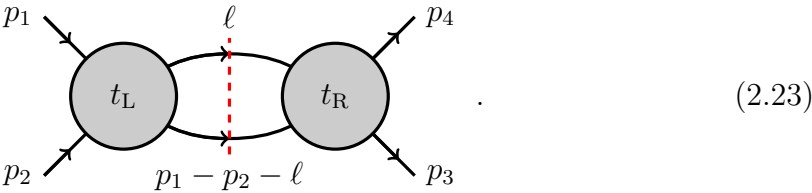

$$\tag{2.23}$$

For the purpose of this section, we took the direction of all the momenta to be from left to right. Assuming that the tree-level amplitude has a Regge growth of $s^{\alpha(t)}$, the Regge growth from such a diagram is schematically

$$\mathrm{Im}\,A \sim \int \mathrm{d}(\text{phase space})\, s^{\alpha(t_{\mathrm{L}})+\alpha(t_{\mathrm{R}})}\,. \tag{2.24}$$

The integral over the on-shell phase space takes the form

$$\int \mathrm{d}^D\ell\,\delta(\ell^2)\,\delta((p_1+p_2-\ell)^2) \sim \frac{1}{\sqrt{|\det\mathcal{G}_{p_1p_2p_3}|}}\int \mathrm{d}t_{\mathrm{L}}\,\mathrm{d}t_{\mathrm{R}}\,\left(-\frac{\det\mathcal{G}_{p_1p_2p_3\ell}}{\det\mathcal{G}_{p_1p_2p_3}}\right)^{\frac{D-5}{2}}\,, \tag{2.25}$$

where D is the space-time dimension. Here we rewrote the integral over the on-shell phase space in terms of an integral over the separate transfer momenta squared

$t_\mathrm{L} = (p_1 - \ell)^2$ and $t_\mathrm{R} = (\ell - p_4)^2$. This introduces various Gram determinants (and trivial numerical factors that we suppressed),

$$\det \mathcal{G}_{p_1 p_2 p_3 \ell} = \det \begin{bmatrix} p_1 \cdot p_1 & p_1 \cdot p_2 & p_1 \cdot p_3 & p_1 \cdot \ell \\ p_1 \cdot p_2 & p_2 \cdot p_2 & p_2 \cdot p_3 & p_2 \cdot \ell \\ p_1 \cdot p_3 & p_2 \cdot p_3 & p_3 \cdot p_3 & p_3 \cdot \ell \\ p_1 \cdot \ell & p_2 \cdot \ell & p_3 \cdot \ell & \ell \cdot \ell \end{bmatrix} . \tag{2.26}$$

All these inner products can be evaluated in terms of the Lorentz invariants $s, t, t_\mathrm{L}, t_\mathrm{R}$ and the masses.

The biggest contribution to the Regge growth of the amplitude comes from the kinematical region for which $t_\mathrm{L}$ and $t_\mathrm{R}$ are maximal. This occurs for the following kinematics that can be taken inside a three-dimensional plane. Take for massless particles

$$p_1 = \frac{\sqrt{s}}{2} \begin{pmatrix} 1 \\ 1 \\ 0 \end{pmatrix} , \quad p_2 = \frac{\sqrt{s}}{2} \begin{pmatrix} 1 \\ -1 \\ 0 \end{pmatrix} , \quad p_3 = \frac{\sqrt{s}}{2} \begin{pmatrix} 1 \\ -\cos\theta \\ -\sin\theta \end{pmatrix} , \quad p_4 = \frac{\sqrt{s}}{2} \begin{pmatrix} 1 \\ \cos\theta \\ \sin\theta \end{pmatrix} . \tag{2.27}$$

Here, $\theta$ is the scattering angle, related to the momentum transfer as $t = -s \sin(\frac{\theta}{2})^2$. The optimal situation occurs when $\ell$ rotates precisely half of the scattering angle, i.e.

$$\ell = \frac{\sqrt{s}}{2} \begin{pmatrix} 1 \\ \cos(\frac{\theta}{2}) \\ \sin(\frac{\theta}{2}) \end{pmatrix} , \tag{2.28}$$

in which case $t_\mathrm{L} = t_\mathrm{R} = -s \sin(\frac{\theta}{4})^2$. In the Regge limit, the scattering angle, becomes very small and we have

$$t_\mathrm{L} = t_\mathrm{R} \sim -\frac{s\theta^2}{16} \sim \frac{t}{4} . \tag{2.29}$$

In other words, the momentum transfer $\sqrt{-t}$ is shared equally between the left and right amplitudes that are glued at the cut.

This implies that to first approximation, the Regge behaviour is given by $s^{2\alpha(\frac{t}{4}) - 1}$, where the additional $-1$ comes from the Gram determinant prefactor in (2.25), which is $\det \mathcal{G}_{p_1 p_2 p_3} = stu \sim s^2$. A more precise estimate also takes into account logarithmic corrections that arise from performing the integral over $t_\mathrm{L}$ and $t_\mathrm{R}$ in the vicinity of the maximum at $t_\mathrm{L} = t_\mathrm{R} = \frac{t}{4}$. This yields up a $t$-dependent prefactor

$$\mathrm{Im}\, A \sim s^{2\alpha(\frac{t}{4}) - 1} \log(s)^{1 - \frac{\mathrm{D}}{2}} . \tag{2.30}$$

In particular, this means that for a linear trajectory of the tree-level amplitudes, $\alpha(t) = J + \alpha' t$, the intercept at one-loop level behaves as $2J - 1$, while the slope halves at one loop.

## 2.6 Reality of one-loop amplitudes

Let us apply the theory now to the one-loop string amplitudes of interest. We refer back to Figure 1 for a summary of the different topologies.

**Closed string.** For the closed string, the amplitude is still crossing symmetric and thus $\mathcal{A}^- = 0$ also at one-loop. The Regge trajectory is $\alpha(t) = 3 + \frac{1}{2}\alpha'_c t$ and thus we have

$$\mathcal{A}_{\text{closed}} \sim \frac{(-i\alpha'_c s)^{3+\frac{1}{2}\alpha'_c t}}{\log(\alpha'_c s)^4} \ , \tag{2.31}$$

up to $s$-independent prefactors and for $s \gg 0$ with slightly positive imaginary part of $s$. We will directly confirm this using saddle-point approximation, see eq. (4.28).

**Open (1342).** This diagram is also invariant under $s$- and $u$-channel (equivalently, $2 \leftrightarrow 3$) exchange since this will reverse the order of the vertex operators under which the open string is invariant. Using that the Regge trajectory is $\alpha(t) = 1 + \frac{1}{2}\alpha' t$, we get

$$\mathcal{A}_{(1342)} \sim \frac{(-i\alpha' s)^{1+\frac{1}{2}\alpha' t}}{\log(\alpha' s)^4} \ , \tag{2.32}$$

up to $s$-independent prefactors. This will again be confirmed by direct saddle-point analysis, see eq. (5.34).

**Open (12)(34) and (13)(24).** These two color structures are exchanged by $s \leftrightarrow u$ crossing. Hence we will be able to predict the reality conditions of their sums and differences. We have

$$\mathcal{A}^+ \sim \frac{(-i\alpha' s)^{1+\frac{1}{2}\alpha' t}}{\log(\alpha' s)^4} \ , \qquad \mathcal{A}^- \sim i\,\frac{(-i\alpha' s)^{1+\frac{1}{2}\alpha' t}}{\log(\alpha' s)^4} \ , \tag{2.33}$$

up to $s$-independent prefactors. This is again exactly consistent with what we find from the saddle-point analysis in eqs. (5.16) and (5.25). Dropping the same $s$-independent prefactors for both amplitudes, we will find from the saddle-point analysis

$$\mathcal{A}_{(12)(34)} \sim i\,\frac{(\alpha' s)^{1+\frac{1}{2}\alpha' t}}{\log(\alpha' s)^4}\, \mathrm{e}^{-\frac{\pi i t}{2}} \ , \qquad \mathcal{A}_{(13)(24)} \sim i\,\frac{(\alpha' s)^{1+\frac{1}{2}\alpha' t}}{\log(\alpha' s)^4} \ . \tag{2.34}$$

Taking the sum and difference confirms the expectation (2.33).

**Open (14)(23).** In this case, we know that there are closed strings propagating in the $t$-channel, which will make the discussion in Section 2.5 breaks down. Instead, the expected Regge trajectory is the tree-level closed string trajectory, $2 + \alpha'_c t$. The diagram is invariant under $s \leftrightarrow u$ crossing and thus we expect

$$\mathcal{A}_{(14)(23)} \sim (-i\alpha' s)^{2+\alpha'_c t} = (-i\alpha' s)^{2+\frac{1}{2}\alpha' t} \ , \tag{2.35}$$

where we used that $\alpha'_c = \frac{1}{2}\alpha'$ for the open string. We will again confirm this by saddle-point approximation, see eq. (5.41).

**Open (1234) and (1324).** These channels are exchanged under $s \leftrightarrow u$ crossing. From field theory considerations, we would predict that

$$\mathcal{A}^+ \sim \frac{(-i\alpha' s)^{1+\frac{1}{2}\alpha' t}}{\log(\alpha' s)^4} , \qquad \mathcal{A}^- \sim i \frac{(-i\alpha' s)^{1+\frac{1}{2}\alpha' t}}{\log(\alpha' s)^4} , \qquad (2.36)$$

as in eq. (2.33). We believe that this is the correct result when $s$ has a positive imaginary part. However, it is much more difficult to extract it from a saddle-point evaluation and likewise from unitarity cuts, which are only valid for real $s$. The reason is that for real $s$, there is a bigger contribution with Regge trajectory $2 + \alpha' t$, which leads to the unexpected behaviour summarized in Figure 1. This contribution is accompanied by a trigonometric factor in $s$ which is exponentially suppressed for small imaginary $s$ and thus not visible as soon as we make $s$ slightly imaginary.

## 3  Baikov representation of the imaginary part

As we shall see, many one-loop amplitudes are purely imaginary in the Regge limit. Thus it is convenient to start discussing the imaginary part of the amplitude, which can – analogously to field theory – be obtained from unitarity cuts of the one-loop amplitude. This will also most directly make contact with field theory. This provides explicit formulas whose Regge limit we will analyze below.

In the following we will set $\alpha' = 1$ and $\alpha'_c = 2$ for readability. This is strictly speaking not consistent when we consider open and closed strings together because $\alpha'_c = \frac{1}{2}\alpha'$. In those cases, we will reinstate the factors of $\alpha'$ and $\alpha'_c$ explicitly.

### 3.1  Closed string

Let us begin with the conceptually simpler case of the closed string amplitude. For the four-point function, the formula for type IIA and IIB strings is identical and takes the form

$$\operatorname{Im} A_{\text{closed}} = \frac{16\pi s^4}{15\sqrt{stu}} \sum_{\sqrt{m_{\text{D}}}+\sqrt{m_{\text{U}}}\leqslant\sqrt{s}} \int \mathrm{d}t_{\text{L}}\, \mathrm{d}t_{\text{R}}\ P_{m_{\text{D}},m_{\text{U}}}^{\frac{5}{2}} Q_{m_{\text{D}},m_{\text{U}}}^2$$
$$\times \frac{\Gamma(-s)\Gamma(-t_{\text{L}})\Gamma(-u_{\text{L}})}{\Gamma(1+s)\Gamma(1+t_{\text{L}})\Gamma(1+u_{\text{L}})} \times \frac{\Gamma(-s)\Gamma(-t_{\text{R}})\Gamma(-u_{\text{R}})}{\Gamma(1+s)\Gamma(1+t_{\text{R}})\Gamma(1+u_{\text{R}})} . \quad (3.1)$$

Here, we suppress the kinematical tensor $t_8\tilde{t}_8$ that keeps track of the polarization structure of the amplitude. We will use the convention that $A$ (as opposed to $\mathcal{A}$) denotes the amplitude with the polarization tensor removed, as well as the colour trace in the case of the open string. For the power counting it is important to remember that $t_8, \tilde{t}_8 \sim s^2$.

This formula comes from a unitarity cut as in field theory, see Figure 2.[1] $m_{\mathrm{D}}$ and $m_{\mathrm{U}}$ label the mass squares of the internal lines (which are integers in string theory). $t_{\mathrm{L}}$ and $t_{\mathrm{R}}$ are the left- and right- momentum transfers, respectively. We have

$$u_{\mathrm{L,R}} = m_{\mathrm{D}} + m_{\mathrm{U}} - s - t_{\mathrm{L,R}} \ . \tag{3.2}$$

The integral over $t_{\mathrm{L}}$ and $t_{\mathrm{R}}$ is the integral over the intermediate on-shell phase space with $P_{m_{\mathrm{D}},m_{\mathrm{U}}}^{\frac{5}{2}}$ the appropriate measure arising from integrating out over the other loop momenta. The integration domain for (3.1) is $P_{m_{\mathrm{D}},m_{\mathrm{D}}} > 0$. Here, $P_{m_{\mathrm{D}},m_{\mathrm{U}}}$ takes the form of a Gram determinant

$$P_{m_{\mathrm{D}},m_{\mathrm{U}}}(s,t,t_{\mathrm{L}},t_{\mathrm{R}}) = -\frac{1}{4stu} \det \begin{bmatrix} 0 & s & u & m_{\mathrm{U}} - s - t_{\mathrm{L}} \\ s & 0 & t & t_{\mathrm{L}} - m_{\mathrm{D}} \\ u & t & 0 & m_{\mathrm{D}} - t_{\mathrm{R}} \\ m_{\mathrm{U}} - s - t_{\mathrm{L}} & t_{\mathrm{L}} - m_{\mathrm{D}} & m_{\mathrm{D}} - t_{\mathrm{R}} & 2m_{\mathrm{D}} \end{bmatrix} . \tag{3.3}$$

This is the explicit from of the Gram determinant (2.26) in this case. Finally, $Q_{m_{\mathrm{D}},m_{\mathrm{U}}}$ are certain polynomials arising from a summation over polarizations of the internal states. They can be defined in terms of a generating function as follows. Let

$$Q_{m_{\mathrm{L}},m_{\mathrm{D}},m_{\mathrm{R}},m_{\mathrm{U}}}(s,t) = [q_{\mathrm{L}}^{m_{\mathrm{L}}} q_{\mathrm{D}}^{m_{\mathrm{D}}} q_{\mathrm{R}}^{m_{\mathrm{R}}} q_{\mathrm{U}}^{m_{\mathrm{U}}}] \prod_{\ell=1}^{\infty} \prod_{a=\mathrm{L,R}} (1 - q^{\ell} q_a^{-1})^{-s} (1 - q^{\ell} q_a)^{-s}$$

$$\times \prod_{a=\mathrm{D,U}} (1 - q^{\ell} q_a^{-1})^{-t} (1 - q^{\ell-1} q_a)^{-t}$$

$$\times \prod_{a=\mathrm{L,R}} (1 - q^{\ell} q_a^{-1} q_{\mathrm{D}}^{-1})^{-u} (1 - q^{\ell-1} q_a q_{\mathrm{D}})^{-u} \ , \tag{3.4}$$

where $q = q_{\mathrm{L}} q_{\mathrm{D}} q_{\mathrm{R}} q_{\mathrm{U}}$ and $[q_{\mathrm{L}}^{m_{\mathrm{L}}} q_{\mathrm{D}}^{m_{\mathrm{D}}} q_{\mathrm{R}}^{m_{\mathrm{R}}} q_{\mathrm{U}}^{m_{\mathrm{U}}}]$ denotes the coefficient of the relevant term in the series expansion around each $q_a = 0$. We then set

$$Q_{m_{\mathrm{D}},m_{\mathrm{U}}}(s,t,t_{\mathrm{L}},t_{\mathrm{R}}) = \sum_{m_{\mathrm{L}},m_{\mathrm{R}}=0}^{m_{\mathrm{D}}+m_{\mathrm{U}}} Q_{m_{\mathrm{L}},m_{\mathrm{D}},m_{\mathrm{R}},m_{\mathrm{U}}}(s,t)(-t_{\mathrm{L}})_{m_{\mathrm{L}}}(-s-t_{\mathrm{L}}+m_{\mathrm{L}}+1)_{m_{\mathrm{D}}+m_{\mathrm{U}}-m_{\mathrm{L}}}$$

$$\times (-t_{\mathrm{R}})_{m_{\mathrm{R}}}(-s - t_{\mathrm{R}} + m_{\mathrm{R}} + 1)_{m_{\mathrm{D}}+m_{\mathrm{U}}-m_{\mathrm{R}}} \ , \tag{3.5}$$

with $(a)_n = a(a+1) \cdots (a+n-1)$ the rising Pochhammer symbol.

This explains all ingredients going into (3.1). The presence of the square of $Q_{m_{\mathrm{D}},m_{\mathrm{U}}}$ is a reflection of the double-copy phenomenon of the closed string, for the open string the corresponding formula that we discuss below does not have a square. The polarization sums can be taken independently for the left- and right-movers

---

[1]Compared to the unitarity cuts descending from the optical theorem, we do not have to complex conjugate the amplitude to the right of the cut. This is because the contributions to the left and right of the cut are both real for a four-point amplitude at one loop.

and hence lead to a square of the polynomials $Q_{m_\mathrm{D},m_\mathrm{U}}$. Since this formula has not appeared in this generality in the literature before, we give a direct derivation of this formula from the worldsheet in Appendix A.

## 3.2 Open string

There is a similar formula for the open type I string amplitudes, which was already stated in [14, 15], but not in all cases of interest.

For the open string, there are a number of different color structures and correspondingly a number of different color contractions in the unitarity cuts. Let us begin with the simplest case: the planar annulus. For it, we have

$$
\mathrm{Im}\ \ \raisebox{-1em}{\includegraphics{fig1}}\ \ = \frac{N\pi s^2}{60\sqrt{stu}} \sum_{\sqrt{m_\mathrm{D}}+\sqrt{m_\mathrm{U}}\leqslant\sqrt{s}} \int \mathrm{d}t_\mathrm{L}\, \mathrm{d}t_\mathrm{R}\ P_{m_\mathrm{D},m_\mathrm{U}}^{\frac{5}{2}} Q_{m_\mathrm{D},m_\mathrm{U}}
$$
$$
\times \frac{\Gamma(-s)\Gamma(-t_\mathrm{L})}{\Gamma(1+u_\mathrm{L})} \times \frac{\Gamma(-s)\Gamma(-t_\mathrm{R})}{\Gamma(1+u_\mathrm{R})} \ . \quad (3.6)
$$

We used a double line notation which indicate the color flow. Here $N = 32$ denotes the size of the gauge group SO(32) of the type I string. It is present since there is an empty color loop in the expression. We suppress the kinematical factor $t_8$ as well as the colour trace $\mathrm{tr}(t^{a_1}t^{a_2}t^{a_3}t^{a_4})$ in this expression. As suggested from field theory, the two amplitudes entering in the unitarity cut are the s-channel massive generalizations of the Veneziano amplitude. Note that there is an equivalent diagram that gives the same value and does not appear separately,

$$
\raisebox{-1em}{\includegraphics{fig2}}\ =\ \raisebox{-1em}{\includegraphics{fig3}}\ \ . \quad (3.7)
$$

It is obtained by sending $(u_\mathrm{L}, u_\mathrm{R}) \leftrightarrow (t_\mathrm{L}, t_\mathrm{R})$ in the above expression. Since both $P_{m_\mathrm{D},m_\mathrm{U}}$ and $Q_{m_\mathrm{D},m_\mathrm{U}}$ are invariant under this expression, the resulting expression is equivalent. We of course also get an identical expression when reflecting the picture along the vertical axis, which amounts to interchanging $(t_\mathrm{L}, u_\mathrm{L}) \longrightarrow (t_\mathrm{R}, u_\mathrm{R})$ and interchanging the particles 12 with the particles 34.

We can now write down the unitarity formula for any type of color contraction by the following rules:

1. Include the color traces corresponding to the boundaries (in particular a factor of $N = 32$ for every empty color loop).

2. Include the correct amplitude for the left and right part in the form

$$\text{(diagram 1,2)} = \frac{\Gamma(-s)\Gamma(-t_{\mathrm{L}})}{\Gamma(1+u_{\mathrm{L}})} \ , \tag{3.8a}$$

$$\text{(diagram 1,2)} = (-1)^{m_{\mathrm{D}}+m_{\mathrm{U}}} \frac{\Gamma(-s)\Gamma(-u_{\mathrm{L}})}{\Gamma(1+t_{\mathrm{L}})} \ , \tag{3.8b}$$

$$\text{(diagram 1,2)} = (-1)^{m_{\mathrm{D}}+m_{\mathrm{U}}} \frac{\Gamma(-t_{\mathrm{L}})\Gamma(-u_{\mathrm{L}})}{\Gamma(1+s)} \ . \tag{3.8c}$$

We refer to these three possibilities as $s$-disk, $t$-disk or $u$-disk. We also call the corresponding unitarity cut and $s-s-$gluing, $t-u-$gluing, etc.

3. Include a factor of $(-1)^{m_{\mathrm{D}}+1}$ when the color lines cross at the bottom part of cut and $(-1)^{m_{\mathrm{U}}+1}$ when they cross at the top part of the cut.

For example, we have

$$\text{Im}\ \text{(diagram 1,2,3,4)} = \frac{\pi s^2}{60\sqrt{stu}} \sum_{\sqrt{m_{\mathrm{D}}}+\sqrt{m_{\mathrm{U}}}\leqslant\sqrt{s}} \int \mathrm{d}t_{\mathrm{L}}\,\mathrm{d}t_{\mathrm{R}}\ P_{m_{\mathrm{D}},m_{\mathrm{U}}}^{\frac{5}{2}} Q_{m_{\mathrm{D}},m_{\mathrm{U}}}$$
$$\times \frac{\Gamma(-s)\Gamma(-t_{\mathrm{L}})}{\Gamma(1+u_{\mathrm{L}})} \times \frac{\Gamma(-t_{\mathrm{R}})\Gamma(-u_{\mathrm{R}})}{\Gamma(1+s)} \ , \tag{3.9}$$

which in our terminology is an $s-u-$gluing with a double crossed color lines. These equations all appeared in some form or another in refs. [14] and [15], but we recall them here all in one place. We give detailed derivations of this formula in all cases in Appendix A.

Even though these formulas just differ by signs, we will see that the physics in the Regge limit is very different in the different cases. We should also note that while a minus sign for a crossing color line is suggested from field theory due to the antisymmetry of the fundamental indices in a $\mathrm{SO}(N)$ representation, the additional factors $(-1)^{m_{\mathrm{D}}}$ and $(-1)^{m_{\mathrm{U}}}$ are less obvious to see.

# 4   Regge behaviour of the closed string

We start by analyzing the Regge behaviour of the closed string amplitude given by eq. (3.1). As we shall see, the sum over $m_{\mathrm{D}}$ and $m_{\mathrm{U}}$ converges absolutely and it hence makes sense to look at the Regge limit of every term in the sum separately. We denote such a term by $\text{Im}\,A_{m_{\mathrm{D}},m_{\mathrm{U}}}$.

## 4.1 Generic momentum transfer

The main simplification expected from Regge theory is that the integral over the on-shell phase space is completely dominated by the Regge limit of the two Virasoro-Shapiro amplitudes entering it [13], see also [19] for a general discussion within field theory. Indeed, there are two dominating regions, near $t_{\mathrm{L}} \sim t_{\mathrm{R}} = \mathcal{O}(1)$ and $u_{\mathrm{L}}, u_{\mathrm{R}} = \mathcal{O}(1)$ as $s \to \infty$. Since the closed string amplitudes are fully crossing symmetric, these regions give two identical contributions and thus we focus on the region where $t_{\mathrm{L}}$ and $t_{\mathrm{R}}$ does not scale with $s$. In particular, we can take the Regge limit of the integrand directly. For $Q_{m_{\mathrm{D}},m_{\mathrm{U}}}$, we use that in the Regge limit

$$Q_{m_{\mathrm{D}},m_{\mathrm{U}}} \sim s^{2m_{\mathrm{D}}+2m_{\mathrm{U}}} \frac{\Gamma(t - t_{\mathrm{L}} - t_{\mathrm{R}} + m_{\mathrm{D}})\Gamma(t - t_{\mathrm{L}} - t_{\mathrm{R}} + m_{\mathrm{U}})}{\Gamma(m_{\mathrm{D}} + 1)\Gamma(m_{\mathrm{U}} + 1)\Gamma(t - t_{\mathrm{L}} - t_{\mathrm{R}})^2} \ . \tag{4.1}$$

This follows directly from the definition (3.4) and (3.5). We demonstrate this identity in Appendix C Combined with the Regge limit of the Virasoro Shapiro amplitudes, we get

$$\operatorname{Im} A_{m_{\mathrm{D}},m_{\mathrm{U}}} \sim \frac{32\pi}{15\sqrt{stu}} \int \mathrm{d}t_{\mathrm{L}} \, \mathrm{d}t_{\mathrm{R}} \ P_{m_{\mathrm{D}},m_{\mathrm{U}}}^{\frac{5}{2}} s^{2(t_{\mathrm{L}}+t_{\mathrm{R}})}$$
$$\times \frac{\Gamma(m_{\mathrm{D}} + t - t_{\mathrm{L}} - t_{\mathrm{R}})^2 \Gamma(m_{\mathrm{U}} + t - t_{\mathrm{L}} - t_{\mathrm{R}})^2}{\Gamma(m_{\mathrm{D}} + 1)^2 \Gamma(m_{\mathrm{U}} + 1)^2 \Gamma(t - t_{\mathrm{L}} - t_{\mathrm{R}})^4}$$
$$\times \frac{\Gamma(-t_{\mathrm{L}}) \sin(\pi(s + t_{\mathrm{L}}))}{\Gamma(1 + t_{\mathrm{L}}) \sin(\pi s)} \times \frac{\Gamma(-t_{\mathrm{R}}) \sin(\pi(s + t_{\mathrm{R}}))}{\Gamma(1 + t_{\mathrm{R}}) \sin(\pi s)} \ . \tag{4.2}$$

In order to get the maximal growth in the $s \to \infty$ regime, we want to maximize $t_{\mathrm{L}} + t_{\mathrm{R}}$, while satisfying the constraint $P_{m_{\mathrm{D}},m_{\mathrm{U}}} \geqslant 0$. In the limit $s \to \infty$, the maximum is attained for

$$t_{\mathrm{L}} = t_{\mathrm{R}} = \frac{t}{4} \ . \tag{4.3}$$

In all regular factors, we can hence put $t_{\mathrm{L}} = t_{\mathrm{R}} = \frac{t}{4}$. This gives

$$\operatorname{Im} A_{m_{\mathrm{D}},m_{\mathrm{U}}} \sim \frac{32\pi}{15s\sqrt{-t}} \int \mathrm{d}t_{\mathrm{L}} \, \mathrm{d}t_{\mathrm{R}} \left( \frac{(t_{\mathrm{L}} - t_{\mathrm{R}})^2}{4t} - \frac{1}{2}\left(t_{\mathrm{L}} + t_{\mathrm{R}} - \frac{t}{2}\right) \right)^{\frac{5}{2}} s^{2(t_{\mathrm{L}}+t_{\mathrm{R}})}$$
$$\times \frac{\Gamma(m_{\mathrm{D}} + \frac{t}{2})^2 \Gamma(m_{\mathrm{U}} + \frac{t}{2})^2 \Gamma(-\frac{t}{4})^2 \sin(\pi(s + \frac{t}{4}))^2}{\Gamma(m_{\mathrm{D}} + 1)^2 \Gamma(m_{\mathrm{U}} + 1)^2 \Gamma(\frac{t}{2})^4 \Gamma(1 + \frac{t}{4})^2 \sin(\pi s)^2} \ . \tag{4.4}$$

The integral over $t_{\mathrm{L}}$ and $t_{\mathrm{R}}$ can now be evaluated and leads to

$$\operatorname{Im} A_{m_{\mathrm{D}},m_{\mathrm{U}}} \sim \frac{\pi^2 \, s^{t-1}}{32 \log(s)^4} \times \frac{\Gamma(m_{\mathrm{D}} + \frac{t}{2})^2 \Gamma(m_{\mathrm{U}} + \frac{t}{2})^2 \Gamma(-\frac{t}{4})^2 \sin(\pi(s + \frac{t}{4}))^2}{\Gamma(m_{\mathrm{D}} + 1)^2 \Gamma(m_{\mathrm{U}} + 1)^2 \Gamma(\frac{t}{2})^4 \Gamma(1 + \frac{t}{4})^2 \sin(\pi s)^2} \ . \tag{4.5}$$

As mentioned above, the sum over $m_{\mathrm{D}}$ and $m_{\mathrm{U}}$ is absolutely convergent and can be performed. In the Regge limit, the upper bound $\sqrt{m_{\mathrm{D}}} + \sqrt{m_{\mathrm{U}}} \leqslant \sqrt{s}$ disappears. In the end we obtain

$$\operatorname{Im} A \sim \frac{\pi^2 \, s^{t-1}}{32 \log(s)^4} \times \frac{\Gamma(1 - t)^2 \Gamma(-\frac{t}{4})^2}{\Gamma(1 - \frac{t}{2})^4 \Gamma(1 + \frac{t}{4})^2} \times \frac{\sin(\pi(s + \frac{t}{4}))^2}{\sin(\pi s)^2} \tag{4.6}$$

We make several comments on the result.

1. (4.6) excludes the contribution from the polarization tensor $t_8 \tilde{t}_8$. The polarization tensor is quartic in the Mandelstams and thus provide another four powers of $s$ to the result, which recovers the quoted result in Figure 1.

2. We only determined the Regge limit of the imaginary part of the amplitude. This gives the correct reality conditions as discussed in Section 2.6 if we make $s$ slightly complex and thus essentially shows that the real part of the amplitude does not contribute in the Regge limit.

3. If we would have performed the computation in supergravity, we would have picked the $m_\mathrm{D} = m_\mathrm{U} = 0$ term and replaced the Virasoro-Shapiro amplitude by $\frac{1}{stu}$. In this case, the integral over $t_\mathrm{L}$ and $t_\mathrm{R}$ is not dominated at the extremal regions $t_\mathrm{L} \sim t_\mathrm{R} = \mathcal{O}(1)$ or $u_\mathrm{L} \sim u_\mathrm{R} = \mathcal{O}(1)$. Because of the exponent $\frac{5}{2} = \frac{D-5}{2}$ in (3.1), the integral is dominated at these two extreme regions for $D \leqslant 5$, while for $D \geqslant 6$, the logic of Regge theory breaks down. Here string theory comes to the rescue in sufficiently high dimensions!

4. (4.6) breaks down in the forward limit where $t = 0$. This is reflected by the fact that the formula has a pole at $t = 0$, even though the actual amplitude does not have a pole. This leads to an extra logarithmic enhancement for $\mathrm{Im}\, A$ that comes exclusively from the $m_\mathrm{D} = m_\mathrm{U} = 0$ term. Adapting the above computation appropriately gives instead

$$\mathrm{Im}\, A\big|_{t=0} \sim \frac{\pi^2}{12 s \log(s)^2} \ . \tag{4.7}$$

## 4.2 Saddle point evaluation

A similar result was obtained by Sundborg in [10] by a direct saddle point approximation of the worldsheet moduli space integral. We give an improved review of his argument.

We start again with the integral representation for the closed string amplitude (4.8):

$$A = \int_{\mathcal{F}} \frac{\mathrm{d}^2 \tau}{(\mathrm{Im}\, \tau)^5} \int_{\mathbb{T}^2} \prod_{j=1}^3 \mathrm{d}^2 z_j \prod_{1 \leqslant i < j \leqslant 4} |\vartheta_1(z_{ij}|\tau)|^{-2 s_{ij}}\, \mathrm{e}^{\frac{2\pi s_{ij} (\mathrm{Im}\, z_{ij})^2}{\mathrm{Im}\, \tau}} \ . \tag{4.8}$$

As observed repeatedly in the literature this integral representation converges for purely imaginary values of the Mandelstam variables $s = s_{12}$ and $t = s_{14}$ [10, 20].

Thus, let us assume that $s$ and $t$ are purely imaginary with $\mathrm{Im}\, s \gg 0$ and large. We put

$$x = z_{32} \ , \qquad y = z_{14} \ , \qquad z = \frac{1}{2}(z_2 - z_1 + z_3 - z_4 + \tau) \ . \tag{4.9}$$

Written in terms of these variables, the closed string amplitude is

$$A = \int_{\mathcal{F}} \frac{\mathrm{d}^2\tau}{\tau_2^5} \int_{\mathbb{T}^2} \mathrm{d}^2 x \, \mathrm{d}^2 y \, \mathrm{d}^2 z \, \exp\left( \frac{4\pi s}{\tau_2} x_2 y_2 + \frac{\pi t((x_2 + y_2)^2 - 4z_2^2)}{\tau_2} \right)$$

$$\times \left| \frac{\vartheta_4(z + \frac{x}{2} + \frac{y}{2}|\tau)\vartheta_4(z - \frac{x}{2} - \frac{y}{2}|\tau)}{\vartheta_4(z + \frac{x}{2} - \frac{y}{2}|\tau)\vartheta_4(z - \frac{x}{2} + \frac{y}{2}|\tau)} \right|^{-2s} \left| \frac{\vartheta_1(x|\tau)\vartheta_1(y|\tau)}{\vartheta_4(z + \frac{x}{2} - \frac{y}{2}|\tau)\vartheta_4(z - \frac{x}{2} + \frac{y}{2}|\tau)} \right|^{-2t} , \quad (4.10)$$

where $x = x_1 + ix_2$ etc. We now consider a regime where

$$1 \ll \tau_2 \ll |s| . \quad (4.11)$$

Indeed, we are interested in the Regge limit and it will be dominated by a region in moduli space where $\tau_2$ is sufficiently large to make contact with the unitarity cuts of field theory. We will see below that $\tau_2$ is of order $\log(|s|)$.

Due to the prefactor $e^{\frac{4\pi s}{\tau_2} x_2 y_2}$, the phase of the integrand is very rapidly oscillating except for the region $x_2 \sim y_2 \sim 0$. This is indeed the saddle point of the expression. Since $|s| \gg |t|$, we only have to care about the phases from the $s$-dependent part, the phase of all the remaining expressions varies slowly in comparison. Thus we obtain

$$\int_{-\infty}^{\infty} \mathrm{d}x_2 \, \mathrm{d}y_2 \, \exp\left( \frac{4\pi s}{\tau_2} x_2 y_2 \right) = \frac{i\tau_2}{2s} , \quad (4.12)$$

and can afterwards put $x_2 = y_2 = 0$ everywhere. We rename $x_1 = x$ and $y_1 = y$. This is the saddle-point localization. The remaining integral becomes

$$A \sim \frac{i}{2s} \int_{\mathcal{F}} \frac{\mathrm{d}^2\tau}{\tau_2^4} \int_0^1 \mathrm{d}x \, \mathrm{d}y \int_{\mathbb{T}^2} \mathrm{d}^2 z \, \exp\left( -\frac{4\pi t z_2^2}{\tau_2} \right)$$

$$\times \left| \frac{\vartheta_4(z + \frac{x}{2} + \frac{y}{2}|\tau)\vartheta_4(z - \frac{x}{2} - \frac{y}{2}|\tau)}{\vartheta_4(z + \frac{x}{2} - \frac{y}{2}|\tau)\vartheta_4(z - \frac{x}{2} + \frac{y}{2}|\tau)} \right|^{-2s} \left| \frac{\vartheta_1(x|\tau)\vartheta_1(y|\tau)}{\vartheta_4(z + \frac{x}{2} - \frac{y}{2}|\tau)\vartheta_4(z - \frac{x}{2} + \frac{y}{2}|\tau)} \right|^{-2t} . \quad (4.13)$$

We next expand the theta-functions to their leading order for large $\tau_2$. For $\vartheta_1$, the leading order is

$$\vartheta_1(x|\tau) \sim q^{\frac{1}{8}} \sin(\pi x) \quad (4.14)$$

and similarly for $\vartheta_1(y|\tau)$. Here we put as usual $q = e^{2\pi i\tau}$. For $\vartheta_4$, the leading order is 1. For the terms raised to the power $-2t$, this approximation is good enough, since we keep $t$ finite. For the terms raised to the power $-2s$, we however need to go to the next order and have

$$\vartheta_4(z|\tau) \sim 1 - 2\cos(2\pi z)q^{\frac{1}{2}} \sim e^{-2\cos(2\pi z)q^{\frac{1}{2}}} . \quad (4.15)$$

This correction will contribute in the regime (4.11). This approximation is good as long as $-\frac{\tau_2}{2} < z_2 < \frac{\tau_2}{2}$, which we choose to be the integration region. We thus obtain

the approximation

$$A \sim \frac{i}{2s} \int_{\tau_2 \gg 1} \frac{\mathrm{d}^2\tau}{\tau_2^4} \int_0^1 \mathrm{d}x \, \mathrm{d}y \int_{\mathbb{T}^2} \mathrm{d}^2z \, |q|^{-\frac{t}{2}} \exp\left(-\frac{4\pi t z_2^2}{\tau_2}\right)$$
$$\times \exp\left(-16s \sin(\pi x)\sin(\pi y) \operatorname{Re} \cos(2\pi z) q^{\frac{1}{2}}\right) \left(4\sin(\pi x)\sin(\pi y)\right)^{-2t} . \quad (4.16)$$

The next step is crucial. We shift

$$\tau_2 \longrightarrow \tau_2 + \frac{1}{\pi} \log\left(\sin(\pi x)\sin(\pi y)(-is)\right) \quad (4.17)$$

After performing this shift, $\tau_2$ is no longer necessarily large and we should thus take the integration region to be the full strip in $\tau$. Similarly, the bounds on $z_2$ were $-\frac{\tau_2}{2} < z_2 < \frac{\tau_2}{2}$, but since we take $\tau_2$ very large, we can extend the integration region all the way to infinity.

Finally, we are interested in the leading order in $s$, which means that in the prefactor $\frac{1}{\tau_2^4}$, as well as in the exponential $\mathrm{e}^{-\frac{4\pi t z_2^2}{\tau_2}}$, we can replace $\tau_2$ by $\frac{1}{\pi}\log(s)$. The exponential then disappears in the large $s$ limit. Thus we obtain

$$A \sim \frac{i\,\pi^4\,(-is)^t}{2s\log^4(s)} \int_0^1 \mathrm{d}x \, \mathrm{d}y \, \mathrm{d}\tau_1 \, \mathrm{d}z \int_{-\infty}^{\infty} \mathrm{d}\tau_2 \, \mathrm{d}z_2 \, |q|^{-\frac{t}{2}} 2^{-4t} \left(\sin(\pi x)\sin(\pi y)\right)^{-t}$$
$$\times \exp\left(-16i \operatorname{Re} \cos(2\pi z) q^{\frac{1}{2}}\right) . \quad (4.18)$$

At this point it only remains to compute the remaining integrals and no more approximations have to be made. To proceed, we set

$$\tau = a + b , \qquad z = \frac{a - b}{2} . \quad (4.19)$$

In terms of these variables,

$$2 \operatorname{Re} \cos(2\pi z) q^{\frac{1}{2}} = \cos(2\pi a_1)\mathrm{e}^{-2\pi a_2} + \cos(2\pi b_1)\mathrm{e}^{-2\pi b_2} . \quad (4.20)$$

Thus this change of variables factorizes the remaining integral and hence

$$A \sim \frac{i\,\pi^4\,(-is)^t\,2^{-4t}}{2s\log^4(s)}$$
$$\times \left[\int_0^1 \mathrm{d}x \, \sin(\pi x)^{-t} \int_0^1 \mathrm{d}a_1 \int_{-\infty}^{\infty} \mathrm{d}a_2 \, \mathrm{e}^{\pi a_2 t} \exp\left(-8i\cos(2\pi a_1)\mathrm{e}^{-2\pi a_2}\right)\right]^2 . \quad (4.21)$$

We have

$$\int_0^1 \mathrm{d}x \, \sin(\pi x)^{-t} = \frac{\Gamma(\frac{1-t}{2})}{\sqrt{\pi}\,\Gamma(1 - \frac{t}{2})} , \quad (4.22)$$

while for the second integral we write

$$u + iv = \mathrm{e}^{2\pi i(a_1 + ia_2)} , \quad (4.23)$$

which brings the integral to the form

$$\frac{1}{4\pi^2} \int du\, dv\, (u^2 + v^2)^{-1-\frac{t}{4}} e^{-8iu} = \frac{\Gamma(\frac{2+t}{4})}{4\pi^{\frac{3}{2}}\Gamma(1+\frac{t}{4})} \int_{-\infty}^{\infty} du\, |u|^{-1-\frac{t}{2}} e^{-8iu} \qquad (4.24)$$

$$= \frac{2^{\frac{3t}{2}}\Gamma(\frac{2+t}{4})\cos(\frac{\pi t}{4})\Gamma(-\frac{t}{2})}{2\pi^{\frac{3}{2}}\Gamma(1+\frac{t}{4})} \qquad (4.25)$$

$$= \frac{2^{t-2}\Gamma(-\frac{t}{4})}{\pi\Gamma(1+\frac{t}{4})} \; . \qquad (4.26)$$

Notice that this integral is essentially the Regge limit of the tree-level Virasoro-Shapiro amplitude. Putting everything together gives

$$A \sim \frac{i\,\pi\,(-is)^t\,2^{-2t}}{32s\log^4(s)} \frac{\Gamma(\frac{1-t}{2})^2\Gamma(-\frac{t}{4})^2}{\Gamma(1-\frac{t}{2})^2\Gamma(1+\frac{t}{4})^2} \qquad (4.27)$$

$$= \frac{i\,\pi^2\,s^t}{32s\log^4(s)} \frac{\Gamma(1-t)^2\Gamma(-\frac{t}{4})^2}{\Gamma(1-\frac{t}{2})^4\Gamma(1+\frac{t}{4})^2} \times e^{-\frac{\pi it}{2}} \; . \qquad (4.28)$$

Sundborg also gives arguments that this result can be analytically continued to any value of $t$ and $\text{Im}\, s > 0$, but without assuming that $s$ is purely imaginary.

This is to be contrasted with $i\,\text{Im}\,A$ given in eq. (4.6), which is valid initially for real large $s$ and real $t < 0$. Clearly, the result is however analytic and we can extend it into the complex plane.[2] Thus, consider putting $\text{Im}\, s > 0$ in eq. (4.6). Then the sine factors further simplify and we have

$$\frac{\sin(\pi(s+\frac{t}{4}))^2}{\sin(\pi s)^2} = \left( \frac{e^{\pi i(s+\frac{t}{4})} - e^{-\pi i(s+\frac{t}{4})}}{e^{\pi is} - e^{-\pi is}} \right)^2 \sim e^{-\frac{\pi it}{2}} \; . \qquad (4.29)$$

Thus for $\text{Im}\, s > 0$, (4.6) becomes identical to (4.28), which in turn matches with the general expectations discussed in Section 2.6.

## 5   Regge behaviour of the open string

As we will see, the Regge behaviour of the open string is quite different. We again start by analyzing the Regge growth of the imaginary diagrams for separate mass-levels and then attempt to sum over all mass-levels.

### 5.1   Contribution from fixed $(m_{\text{D}}, m_{\text{U}})$

Let us write the contribution to the imaginary part from the mass-level $(m_{\text{D}}, m_{\text{U}})$ of a diagram as

$$\text{Im}\, A_{m_{\text{D}}, m_{\text{U}}} = \frac{\pi s^2}{60\sqrt{stu}} \int dt_{\text{L}}\, dt_{\text{R}}\; P_{m_{\text{D}}, m_{\text{U}}}^{\frac{5}{2}} Q_{m_{\text{D}}, m_{\text{U}}} A_{\text{L}}(s, t_{\text{L}}) A_{\text{R}}(s, t_{\text{R}}) \; , \qquad (5.1)$$

---

[2]It is not obvious that this is allowed. Limits and analytic continuation do not have to commute in general.

where both $A_\mathrm{L}$ and $A_\mathrm{R}$ are either the $s$-, $t$- or $u$-disk amplitude as in eq. (3.8). There is also a possible overall sign determined by the crossing of the lines which we insert later.

As for the closed string, the integral is dominated from the two regions $t_\mathrm{L} \sim t_\mathrm{R} \sim \mathcal{O}(1)$ and $u_\mathrm{L} \sim u_\mathrm{R} \sim \mathcal{O}(1)$. They are precisely related by the flipping operation (3.7). Thus, we can restrict our attention to the region with $t_\mathrm{L} \sim t_\mathrm{R} \sim \mathcal{O}(1)$ and for the Regge analysis count the two diagrams (3.7) as different. One can thus input the Regge growth of $A_\mathrm{L}$ and $A_\mathrm{R}$. Regardless of the channel, the amplitudes behave like

$$A_\mathrm{L}(s, t_\mathrm{L}) \sim s^{t_\mathrm{L} - 1 - m_\mathrm{D} - m_\mathrm{U}} A_\mathrm{L}^{\mathrm{Regge}}(s, t_\mathrm{L}) , \tag{5.2}$$

where the remaining piece is of order one in $s$. Using also (4.1) gives

$$\mathrm{Im}\, A_{m_\mathrm{D}, m_\mathrm{U}} \sim \frac{\pi}{60\sqrt{stu}} \int \mathrm{d}t_\mathrm{L}\, \mathrm{d}t_\mathrm{R}\; P_{m_\mathrm{D}, m_\mathrm{U}}^{\frac{5}{2}} s^{t_\mathrm{L} + t_\mathrm{R}}$$
$$\times \frac{\Gamma(t - t_\mathrm{L} - t_\mathrm{R} + m_\mathrm{D})\Gamma(t - t_\mathrm{L} - t_\mathrm{R} + m_\mathrm{U})}{\Gamma(m_\mathrm{D} + 1)\Gamma(m_\mathrm{U} + 1)\Gamma(t - t_\mathrm{L} - t_\mathrm{R})^2} A_\mathrm{L}^{\mathrm{Regge}}(s, t_\mathrm{L}) A_\mathrm{R}^{\mathrm{Regge}}(s, t_\mathrm{R}) . \tag{5.3}$$

Maximizing the exponent of $s$ leads again to the conclusion that the $t_\mathrm{L}$ and $t_\mathrm{R}$ is close to $\frac{t}{4}$ as in the case of the closed string. We thus may put $t_\mathrm{L} = t_\mathrm{R} = \frac{t}{4}$ in all regular expressions. Expanding also $P_{m_\mathrm{D}, m_\mathrm{U}}$ around that point leads to

$$\mathrm{Im}\, A_{m_\mathrm{D}, m_\mathrm{U}} \sim \frac{\pi s^{\frac{t}{2} - 1}}{60\sqrt{-t}} \frac{\Gamma(\frac{t}{2} + m_\mathrm{D})\Gamma(\frac{t}{2} + m_\mathrm{U})}{\Gamma(m_\mathrm{D} + 1)\Gamma(m_\mathrm{U} + 1)\Gamma(\frac{t}{2})^2} A_\mathrm{L}^{\mathrm{Regge}}(s, \tfrac{t}{4}) A_\mathrm{R}^{\mathrm{Regge}}(s, \tfrac{t}{4})$$
$$\times \int \mathrm{d}t_\mathrm{L}\, \mathrm{d}t_\mathrm{R} \left( \frac{(t_\mathrm{L} - t_\mathrm{R})^2}{4t} - \frac{1}{2}\left(t_\mathrm{L} + t_\mathrm{R} - \tfrac{t}{2}\right) \right)^{\frac{5}{2}} s^{t_\mathrm{L} + t_\mathrm{R} - \frac{t}{2}} \tag{5.4}$$
$$= \frac{\pi^2 s^{\frac{t}{2} - 1}}{256 \log(s)^4} \frac{\Gamma(\frac{t}{2} + m_\mathrm{D})\Gamma(\frac{t}{2} + m_\mathrm{U})}{\Gamma(m_\mathrm{D} + 1)\Gamma(m_\mathrm{U} + 1)\Gamma(\frac{t}{2})^2} A_\mathrm{L}^{\mathrm{Regge}}(s, \tfrac{t}{4}) A_\mathrm{R}^{\mathrm{Regge}}(s, \tfrac{t}{4}) . \tag{5.5}$$

It is finally easy to check that $A_\mathrm{L}^{\mathrm{Regge}}(s, \tfrac{t}{4})$ depends on $m_\mathrm{D}$ and $m_\mathrm{U}$ only via an overall phase $(-1)^{m_\mathrm{D} + m_\mathrm{U}}$ independent of which channel we pick. Thus, including the overall sign coming from the crossing of lines, we get

$$\mathrm{Im}\, A_{m_\mathrm{D}, m_\mathrm{U}} \sim (-1)^{c_\mathrm{U}(m_\mathrm{U} + 1) + c_\mathrm{D}(m_\mathrm{D} + 1)} \frac{\pi^2 s^{\frac{t}{2} - 1}}{256 \log(s)^4} \frac{\Gamma(\frac{t}{2} + m_\mathrm{D})\Gamma(\frac{t}{2} + m_\mathrm{U})\Gamma(-\frac{t}{4})^2}{\Gamma(m_\mathrm{D} + 1)\Gamma(m_\mathrm{U} + 1)\Gamma(\frac{t}{2})^2}$$
$$\times \begin{cases} -\frac{\sin(\pi(s + \frac{t}{4}))}{\sin(\pi s)} & s\text{-disk} \\ \frac{\sin(\frac{\pi t}{4})}{\sin(\pi s)} & t\text{-disk} \\ 1 & u\text{-disk} \end{cases} \times \begin{cases} -\frac{\sin(\pi(s + \frac{t}{4}))}{\sin(\pi s)} & s\text{-disk} \\ \frac{\sin(\frac{\pi t}{4})}{\sin(\pi s)} & t\text{-disk} \\ 1 & u\text{-disk} \end{cases} . \tag{5.6}$$

Here $c_\mathrm{D,U} \in \{0, 1\}$ specify the number of times the lines cross at the bottom or top of the cutting diagram. We verified numerically that the exact $\mathrm{Im}\, A_{m_\mathrm{D}, m_\mathrm{U}}$ indeed approaches (5.6) for very large values of $s \gtrsim 10^{100}$, see Figure 3. Since corrections are suppressed by powers of $\log(s)$, convergence is extremely slow.

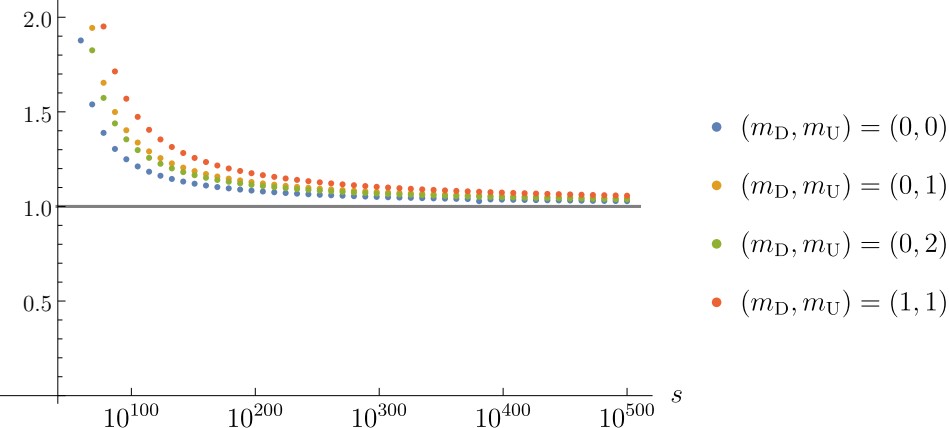

**Figure 3:** Plot of the ratio of (5.6) to the exact value Im $A_{m_\mathrm{D},m_\mathrm{U}}(s, t = -\frac{1}{2})$ for large energies $s$. For illustration, we chose the planar contribution ($c_\mathrm{U} = c_\mathrm{D} = 0$) in the $s$-channel and sampled $s = 10^{10k} + \frac{1}{2}$ for integer $k$.

## 5.2   Regge attenuation: Summing over $m_\mathrm{D}$ and $m_\mathrm{U}$

We would now think that we can obtain the Regge limit of the amplitude by summing over $m_\mathrm{D}$ and $m_\mathrm{U}$. However, as we shall see below that is some cases somewhat premature since to conclude this, we need to exchange the limit $s \to \infty$ with the sum over $m_\mathrm{D}$ and $m_\mathrm{U}$, which becomes infinite in the limit $s \to \infty$. We have

$$\sum_{m=0}^{\infty} (-1)^{c(m+1)} \frac{\Gamma(\frac{t}{2} + m)}{\Gamma(m+1)\Gamma(\frac{t}{2})} = \sum_{m=0}^{\infty} \frac{\Gamma(1 - \frac{t}{2})(-1)^{m+c(m+1)}}{\Gamma(m+1)\Gamma(1 - m - \frac{t}{2})} \tag{5.7}$$

$$= (-1)^c \sum_{m=0}^{\infty} \binom{-\frac{t}{2}}{m} x^m \bigg|_{x=(-1)^{1+c}} \tag{5.8}$$

$$= (-1)^c (1+x)^{-\frac{t}{2}} \big|_{x=(-1)^{1+c}} \tag{5.9}$$

$$= \begin{cases} 0 & c = 0 , \\ -2^{-\frac{t}{2}} & c = 1 . \end{cases} \tag{5.10}$$

Thus, to get something non-zero in the Regge limit, we need the color lines of the diagram to cross both at the top and the bottom. Otherwise, there are cancellations between the infinite tower of states running in the loop. In other words, a naive Regge analysis based on any finite subset of diagrams (for example, by truncating the spectrum of the states going through the cut at some large mass) would fail. We call this effect Regge attenuation.

In the double crossed case, we thus conclude for the imaginary part

$$\mathrm{Im}\, A\big|_{\text{double-crossed}} = \frac{\pi^2 2^{-t} s^{\frac{t}{2}-1} \Gamma(-\frac{t}{4})^2}{256 \log(s)^4} \begin{cases} -\frac{\sin(\pi(s+\frac{t}{4}))}{\sin(\pi s)} & s\text{-disk} \\ \frac{\sin(\frac{\pi t}{4})}{\sin(\pi s)} & t\text{-disk} \\ 1 & u\text{-disk} \end{cases}$$

$$\times \begin{cases} -\frac{\sin(\pi(s+\frac{t}{4}))}{\sin(\pi s)} & s\text{-disk} \\[2mm] \frac{\sin(\frac{\pi t}{4})}{\sin(\pi s)} & t\text{-disk} \\[2mm] 1 & u\text{-disk} \end{cases} \quad . \quad (5.11)$$

We will confirm this behaviour below from an explicit saddle-point computation as in the closed string, where we study the limit of large imaginary $s$. However, this will only be possible for the diagrams involving the $s$- and $u$-disk expression, since the trigonometric factor of the diagram involving the $t$-disk decays exponentially for large values of $\mathrm{Im}\, s$ and is thus only leading for real values of $s$ in the Regge limit.

## 5.3  Saddle-point evaluation for the double-crossed case

We now show that the saddle-point evaluation of the closed string amplitude extends straightforwardly to the open string cases. In all cases, we find results consistent with the general expectation discussed in Section 2.6

**Non-planar annulus $s - s-$gluing.** We start with the diagram

$$(5.12)$$

The corresponding string amplitude is given by the worldsheet integral

$$A^{\text{n-p}} = \frac{1}{64} \int_0^\infty \mathrm{d}\tau_2 \int \mathrm{d}z_1\, \mathrm{d}z_2\, \mathrm{d}z_3 \ \left( \frac{\vartheta_1(z_{21}, i\tau_2)\vartheta_1(z_{43}, i\tau_2)}{\vartheta_4(z_{31}, i\tau_2)\vartheta_4(z_{42}, i\tau_2)} \right)^{-s}$$
$$\times \left( \frac{\vartheta_4(z_{32}, i\tau_2)\vartheta_4(z_{41}, i\tau_2)}{\vartheta_4(z_{31}, i\tau_2)\vartheta_4(z_{42}, i\tau_2)} \right)^{-t} \quad . \quad (5.13)$$

The prefactor $\frac{1}{64}$ arises as follows. We have an explicit prefactor of $\frac{1}{32} = \frac{1}{N}$ due to our normalization conventions in which the planar annulus amplitude is normalized with a prefactor 1. Another factor of $\frac{1}{2}$ is present because this color ordering is automatically invariant under orientation reversal and orientation reversal is gauged in type I string theory.

We are again assuming that $s$ and $t$ are purely imaginary so that the integral converges and want to evaluate the integral for large $\mathrm{Im}\, s$.

Analogously to the closed string, we will focus on the regime

$$1 \ll \tau_2^{-1} \ll |s| \ , \quad (5.14)$$

since in this parametrization, the unitarity cuts come from the region $\tau_2 \to 0$. Thus it is convenient to first perform a modular transformation on the integrand. We also set

$$x = z_{32}\tau_2 \ , \qquad y = z_{14}\tau_2 \ , \qquad z = \frac{\tau_2}{2}(z_2 - z_1 + z_3 - z_4 + 1) \ . \quad (5.15)$$

After renaming $\tau_2 \to \frac{1}{\tau_2}$ and using the modular properties of the theta-functions, this brings the amplitude into the form

$$A^{\text{n-p}} = \frac{1}{64} \int \frac{d\tau_2}{\tau_2^5} \int dx\, dy\, dz \ \exp\left( \frac{2\pi sxy}{\tau_2} + \frac{\pi t}{2\tau_2}\left((x+y)^2 - 4z^2\right) \right)$$
$$\times \left( \frac{\vartheta_4(i(z+\frac{x}{2}+\frac{y}{2}), i\tau_2)\vartheta_4(i(z-\frac{x}{2}-\frac{y}{2}), i\tau_2)}{\vartheta_3(i(z+\frac{x}{2}-\frac{y}{2}), i\tau_2)\vartheta_3(i(z-\frac{x}{2}+\frac{y}{2}), i\tau_2)} \right)^{-s}$$
$$\times \left( \frac{\vartheta_2(ix, i\tau_2)\vartheta_2(iy, i\tau_2)}{\vartheta_3(i(z+\frac{x}{2}-\frac{y}{2}), i\tau_2)\vartheta_3(i(z-\frac{x}{2}+\frac{y}{2}), i\tau_2)} \right)^{-t} . \tag{5.16}$$

We now proceed as before. We notice that the prefactor $e^{\frac{2\pi sxy}{\tau_2}}$ oscillates very fast except when $x \sim y \sim 0$. Thus we evaluate the integral over them via saddle-point evaluation and put afterwards $x = y = 0$ everywhere else. This leads to

$$A^{\text{n-p}} \sim \frac{i}{64s} \int \frac{d\tau_2}{\tau_2^4} \int dz \ e^{-\frac{2\pi tz^2}{\tau_2}} \left( \frac{\vartheta_4(iz, i\tau_2)}{\vartheta_3(iz, i\tau_2)} \right)^{-2s} \left( \frac{\vartheta_2(0, i\tau_2)}{\vartheta_3(iz, i\tau_2)} \right)^{-2t} \tag{5.17}$$

$$\sim \frac{2^{-2t}}{64|s|} \int \frac{d\tau_2}{\tau_2^4} \int dz \ e^{-\frac{2\pi tz^2}{\tau_2}} \ q^{-\frac{t}{4}} \ e^{8sq^{\frac{1}{2}} \cosh(2\pi z)} . \tag{5.18}$$

We then shift

$$\tau_2 \to \tau_2 + \frac{1}{\pi} \log|s| , \tag{5.19}$$

which to leading order in $s$ produces

$$A^{\text{n-p}} \sim \frac{2^{-2t}\pi^4 |s|^{\frac{t}{2}-1}}{64 \log(s)^4} \int_{-\infty}^{\infty} d\tau_2\, dz \ q^{-\frac{t}{4}} e^{8iq^{\frac{1}{2}} \cosh(2\pi z)} \tag{5.20}$$

$$= \frac{2^{-2t}\pi^4 |s|^{\frac{t}{2}-1}}{64 \log(s)^4} \left[ \int_{-\infty}^{\infty} da \ e^{\frac{\pi at}{2} + 4ie^{-2\pi a}} \right]^2 \tag{5.21}$$

$$= \frac{2^{-t}\pi^2 \, i \, s^{\frac{t}{2}-1} e^{-\frac{\pi it}{2}} \, \Gamma(-\frac{t}{4})^2}{256 \log(s)^4} , \tag{5.22}$$

where we factorized the integrand by putting

$$\tau_2 = a + b , \qquad z = \frac{a-b}{2} , \tag{5.23}$$

and the remaining integral becomes the usual defining integral of the Gamma function.

This matches with (5.11) when we take $s$ to be in the upper half plane and confirms in particular that the amplitude is purely imaginary in the Regge limit.

**Non-planar annulus $u - u-$gluing.** We next consider the diagram

$$\tag{5.24}$$

The corresponding string amplitude is given by the analogous expression to (5.16), except that some theta-functions are different, reflecting the fact that vertex operators are on different boundaries. We obtain

$$A^{\text{n-p}} = \frac{1}{64} \int \frac{d\tau_2}{\tau_2^5} \int dx \, dy \, dz \ \exp\left( \frac{2\pi s x y}{\tau_2} + \frac{\pi t}{2\tau_2}\big((x+y)^2 - 4z^2\big) \right)$$
$$\times \left( \frac{\vartheta_3(i(z + \frac{x}{2} + \frac{y}{2}), i\tau_2)\vartheta_3(i(z - \frac{x}{2} - \frac{y}{2}), i\tau_2)}{\vartheta_4(i(z + \frac{x}{2} - \frac{y}{2}), i\tau_2)\vartheta_4(i(z - \frac{x}{2} + \frac{y}{2}), i\tau_2)} \right)^{-s}$$
$$\times \left( \frac{\vartheta_2(ix, i\tau_2)\vartheta_2(iy, i\tau_2)}{\vartheta_4(i(z + \frac{x}{2} - \frac{y}{2}), i\tau_2)\vartheta_4(i(z - \frac{x}{2} + \frac{y}{2}), i\tau_2)} \right)^{-t} . \tag{5.25}$$

We can proceed as before. Only one sign in the exponent changes and we obtain

$$A^{\text{n-p}} \sim \frac{2^{-2t}\pi^4 \, |s|^{\frac{t}{2}-1}}{64 \log(s)^4} \left[ \int_{-\infty}^{\infty} da \ e^{\frac{\pi a t}{2} - 4ie^{-2\pi a}} \right]^2 = \frac{2^{-t}\pi^2 \, i \, s^{\frac{t}{2}-1} \, \Gamma(-\frac{t}{4})^2}{256 \log(s)^4} , \tag{5.26}$$

which again matches with (5.11) and shows that the amplitude becomes purely imaginary in this limit.

**Möbius strip $s - u-$gluing.** Let us discuss the third possibility double-crossed possibility that we can check from the saddle-point analysis given by the following Möbius strip diagram

$$\tag{5.27}$$

The full Möbius strip amplitude can be written as

$$-\int_{\frac{1}{2}+i\mathbb{R}_{\geqslant 0}} (-i \, d\tau) \int \prod_j dz_j \left( \frac{\vartheta_1(z_{21}, \tau)\vartheta_1(z_{34}, \tau)}{\vartheta_1(z_{13}, \tau)\vartheta_4(z_{24}, \tau)} \right)^{-s} \left( \frac{\vartheta_1(z_{32}, \tau)\vartheta_1(z_{14}, \tau)}{\vartheta_1(z_{13}, \tau)\vartheta_1(z_{24}, \tau)} \right)^{-t} , \tag{5.28}$$

where the integration region is $0 = z_4 < z_3 < z_1 < z_2 < 1$. To bring this into a similar form as the other diagrams, we perform a modular transformation in $\tau$. This maps the region close to $\tau = \frac{1}{2}$ to values close to the imaginary axis with large imaginary part, which we can hence write as $i\tau_2$. We also write

$$x = (2z_{32} - 1)\tau_2 , \quad y = (2z_{14} - 1)\tau_2 , \quad z = (z_2 - z_1 + z_3 - z_4 - \tfrac{1}{2})\tau_2 . \tag{5.29}$$

Indeed, for the diagram (5.27), we expect to get an appreciable contribution for $z_{23} \sim \frac{1}{2}$ and $z_{14} \sim \frac{1}{2}$, since for a thin Moebius strip, the corresponding two vertex operators will be close together, but on opposite boundaries of the Möbius strip.

Taking into account the Jacobians from the transformation, we obtain

$$
A^{\mathrm{M}} = -\frac{1}{32} \int \frac{\mathrm{d}\tau_2}{\tau_2^5} \int \mathrm{d}x\,\mathrm{d}y\,\mathrm{d}z \ \exp\left(\frac{2\pi s x y}{\tau_2} + \frac{\pi t}{2\tau_2}\left((x+y)^2 - 4z^2\right)\right)
$$
$$
\times \left(\frac{\vartheta_4(i(z+\frac{x}{2}+\frac{y}{2})+\frac{1}{4}, i\tau_2 - \frac{1}{2})\vartheta_4(i(z-\frac{x}{2}-\frac{y}{2})+\frac{1}{4}, i\tau_2 - \frac{1}{2})}{\vartheta_4(i(z+\frac{x}{2}-\frac{y}{2})-\frac{1}{4}, i\tau_2 - \frac{1}{2})\vartheta_4(i(z-\frac{x}{2}+\frac{y}{2})-\frac{1}{4}, i\tau_2 - \frac{1}{2})}\right)^{-s}
$$
$$
\times \left(\frac{e^{\frac{\pi i}{4}}\vartheta_1(ix+\frac{1}{2}, i\tau_2 - \frac{1}{2})\vartheta_1(iy+\frac{1}{2}, i\tau_2 - \frac{1}{2})}{\vartheta_4(i(z+\frac{x}{2}-\frac{y}{2})-\frac{1}{4}, i\tau_2 - \frac{1}{2})\vartheta_4(i(z-\frac{x}{2}+\frac{y}{2})-\frac{1}{4}, i\tau_2 - \frac{1}{2})}\right)^{-t} \ . \tag{5.30}
$$

We see that there is indeed a saddle-point at $x = y = 0$ and we may compute the integral over $x$ and $y$ as before. This leads to

$$
A^{\mathrm{M}} \sim -\frac{i}{32s} \int \frac{\mathrm{d}\tau_2}{\tau_2^4} \int \mathrm{d}z \ e^{-\frac{2\pi t z^2}{\tau_2}} \left(\frac{\vartheta_4(iz+\frac{1}{4}, i\tau_2 - \frac{1}{2})}{\vartheta_4(iz-\frac{1}{4}, i\tau_2 - \frac{1}{2})}\right)^{-2s} \left(\frac{\vartheta_1(\frac{1}{2}, i\tau_2 - \frac{1}{2})}{\vartheta_4(iz-\frac{1}{4}, i\tau_2 - \frac{1}{2})}\right)^{-2t}
$$
$$
\tag{5.31}
$$

$$
\sim -\frac{i\,2^{-2t}}{32s} \int \frac{\mathrm{d}\tau_2}{\tau_2^4} \int \mathrm{d}z \ q^{-\frac{t}{4}}\, e^{-\frac{2\pi t z^2}{\tau_2}}\, e^{-8sq^{\frac{1}{2}}\sinh(2\pi z)} \ . \tag{5.32}
$$

We then perform the by now familiar manipulations of shifting $\tau_2$ as in (5.19) and changing the variables as in (5.23) to obtain to leading order in large $s$,

$$
A^{\mathrm{M}} \sim \frac{2^{-2t}\pi^4(-is)^{\frac{t}{2}-1}}{32\log(s)^4} \int_{-\infty}^{\infty} \mathrm{d}a \ e^{\frac{\pi a t}{2} + 4ie^{-2\pi a}} \int_{-\infty}^{\infty} \mathrm{d}b \ e^{\frac{\pi b t}{2} - 4ie^{-2\pi b}} \tag{5.33}
$$

$$
= \frac{2^{-t}\pi^2\, i\, s^{\frac{t}{2}-1} e^{-\frac{\pi i t}{4}}\Gamma(-\frac{t}{4})^2}{128\log(s)^4} \ . \tag{5.34}
$$

This again matches the prediction from (5.11) from the $s - u$- and $u - s$-channel together.

## 5.4 Graviton exchange in the non-planar diagram

Let us now consider the diagram

$$
\tag{5.35}
$$

The corresponding amplitude reads

$$
A^{\mathrm{P}} = \int \mathrm{d}\tau_2\,\mathrm{d}x\,\mathrm{d}y\,\mathrm{d}z \ \left(\frac{\vartheta_3(z+\frac{x}{2}+\frac{y}{2}|i\tau_2)\vartheta_3(z-\frac{x}{2}-\frac{y}{2}|i\tau_2)}{\vartheta_3(z+\frac{x}{2}-\frac{y}{2}|i\tau_2)\vartheta_3(z-\frac{x}{2}+\frac{y}{2}|i\tau_2)}\right)^{-s}
$$
$$
\times \left(\frac{\vartheta_1(x|i\tau_2)\vartheta_1(y|i\tau_2)}{\vartheta_3(z+\frac{x}{2}-\frac{y}{2}|i\tau_2)\vartheta_3(z-\frac{x}{2}+\frac{y}{2}|i\tau_2)}\right)^{-t} \ . \tag{5.36}
$$

We run into a problem if we try to replicate the saddle point evaluation that we discussed above for the other open string diagrams. We find since the first factor in (5.36) is exactly unity for $x = y = 0$, the saddle-point integral is not stabilized in the $\tau_2$ direction.

Thereason is that the saddle point is located at the opposite side of the moduli space, corresponding to the closed string degeneration. This is the region $\tau_2 \to \infty$ in this parametrization. Notice that this is opposite from the parametrization used in (5.16), (5.25) and (5.27) in which $\tau_2 \to \frac{1}{\tau_2}$.

Let us analyze the integral more carefully in that region. Expanding $\vartheta_3$ to first order in $\tau_2$ leads to the expression

$$A^{\text{n-p}} \sim \int \mathrm{d}\tau_2 \, \mathrm{d}x \, \mathrm{d}y \, \mathrm{d}z \; e^{8sq^{\frac{1}{2}} \sin(\pi x)\sin(\pi y)\cos(2\pi z)} \big( 4q^{\frac{1}{4}} \sin(\pi x)\sin(\pi y) \big)^{-t} , \qquad (5.37)$$

where $q = e^{-2\pi\tau_2}$ (this is different from what we had above since we are considering the modular transformed $\tau_2$). As we can see, for large values of $\tau_2$ such that $sq^{\frac{1}{2}} \sim 1$, the saddle-point evaluation of the $x$ and $y$ integral breaks down. Instead, we shift

$$\tau_2 \to \tau_2 + \frac{1}{\pi} \log(-is) + \frac{1}{\pi} \log \big( \sin(\pi x)\sin(\pi y) \big) , \qquad (5.38)$$

where we assume again that $s$ is purely imaginary. After this $\tau_2$ is no longer necessarily large and has to be integrated over the whole real line. We end up with

$$A^{\text{n-p}} \sim 4^{-t}(-is)^{\frac{t}{2}} \int_{-\infty}^{\infty} \mathrm{d}\tau_2 \int_{-\frac{1}{2}}^{\frac{1}{2}} \mathrm{d}z \; q^{-\frac{t}{4}} e^{8iq^{\frac{1}{2}}\cos(2\pi z)} \left[ \int_0^1 \mathrm{d}x \; \sin(\pi x)^{-\frac{t}{2}} \right]^2 \qquad (5.39)$$

$$= \frac{2^{-t}(-is)^{\frac{t}{2}} \Gamma(\frac{1}{2} - \frac{t}{4})^2 \Gamma(-\frac{t}{4})}{2\pi^2 \, \Gamma(1 - \frac{t}{4})^2 \Gamma(1 + \frac{t}{4})} \qquad (5.40)$$

$$= -\frac{2^{3-t}(-is)^{\frac{t}{2}} \Gamma(\frac{1}{2} - \frac{t}{4})^2 \sin(\frac{\pi t}{4})}{\pi^3 t^2} , \qquad (5.41)$$

where we used that we evaluated the relevant integrals already in the closed string Section 4.2 and simplified the result.

This contribution originates from the tree-level exchange of Reggeized gravitons in the dual channel. We can read off that the Regge intercept is 2 (taking into account the contribution from the $t_8$-tensor), the Regge slope is $\alpha_c' = \frac{1}{2}\alpha' = \frac{1}{2}$ in our conventions and the absence of logarithmic factors indicates that this is a tree-level contribution. Due to the higher Regge intercept, this contribution is leading over the field theory expectation.

As it stands, this contribution is neither real nor imaginary when we analytically continue back to real $s$ and thus has to be dressed with trigonometric factors. If we would have assumed $\text{Im}(s) < 0$ in the above derivation, we would have found the opposite sign of $i$. We can essentially uniquely guess the correct prefactor by noticing

that the full diagram should also have $s$-channel poles and thus we suspect that

$$
e^{-\frac{\pi it}{4}} \longrightarrow \frac{\sin(\pi(s + \frac{t}{4}))}{\sin(\pi s)} \ ,
\tag{5.42}
$$

which leads to the correct phase. However, as discussed further in Section 7.1, this should be taken with a grain of salt. To summarize, we found the following *real* leading asymptotics in the Regge limit:

$$
A^{\text{n-p}} \sim \frac{2^{-t} s^{\frac{t}{2}} \Gamma(\frac{1}{2} - \frac{t}{4})^2 \Gamma(-\frac{t}{4})}{8\pi^2 \, \Gamma(1 - \frac{t}{4})^2 \Gamma(1 + \frac{t}{4})} e^{-\frac{\pi it}{4}}
\tag{5.43}
$$

for $s$ in the upper half-plane and we suspect that its limit on the real axis is given by (5.42).

# 6 Planar annulus and genus resummation

We will finally address the perhaps most interesting diagram – the planar contribution to the amplitude in the color traces $\text{tr}(t^{a_1} t^{a_2} t^{a_3} t^{a_4})$. We first discuss the one-loop evaluation as above.

## 6.1 The situation at one loop

The planar annulus amplitude in similar variables as above takes the form

$$
A^{\text{p}} = \int \frac{\mathrm{d}\tau_2}{\tau_2^5} \int \mathrm{d}x \, \mathrm{d}y \, \mathrm{d}z \ \exp\left( \frac{2\pi s x y}{\tau_2} + \frac{\pi t}{2\tau_2}\left( (x+y)^2 - 4z^2 \right) \right)
$$
$$
\times \left( \frac{\vartheta_1(i(z + \frac{x}{2} + \frac{y}{2}), i\tau_2)\vartheta_1(i(z - \frac{x}{2} - \frac{y}{2}), i\tau_2)}{\vartheta_1(i(z + \frac{x}{2} - \frac{y}{2}), i\tau_2)\vartheta_1(i(z - \frac{x}{2} + \frac{y}{2}), i\tau_2)} \right)^{-s}
$$
$$
\times \left( \frac{\vartheta_1(ix, i\tau_2)\vartheta_1(iy, i\tau_2)}{\vartheta_1(i(z + \frac{x}{2} - \frac{y}{2}), i\tau_2)\vartheta_1(i(z - \frac{x}{2} + \frac{y}{2}), i\tau_2)} \right)^{-t} .
\tag{6.1}
$$

We have to impose the additional constraint $x \geqslant 0$ and $y \geqslant 0$ originating from the planar ordering of vertex operators. We again see that for large $s$, this integral is dominated from the region $x \sim y \sim 0$, since otherwise the phase oscillates very fast. The local behaviour near those points is however quite different. We have the integral

$$
\int_{x>0, y>0, x^2+y^2<1} \mathrm{d}x \, \mathrm{d}y \ \exp\left( A s x y \right) x^{-t} y^{-t} \sim (-sA)^{t-1} \Gamma(1-t) \log(s) .
\tag{6.2}
$$

This indicates that the leading large $s$ growth of the amplitude is $s^{t-1} \log(s)$, which corresponds to a term that corrects the Regge slope at one loop. However, we will see below that this is not the full picture. We also see that the remaining integral is independent of $s$ (contrary to what happened before). Thus, there is no sense in which the integral is stabilized at large values of $\tau_2$.

**$s^{t-1}\log(s)$ contribution.** When computing this contribution, it is more convenient to use the original parametrization in which we perform a modular transformation $\tau_2 \to \frac{1}{\tau_2}$. We obtain

$$A^{\mathrm{p}} = \int \mathrm{d}\tau_2 \, \mathrm{d}x \, \mathrm{d}y \, \mathrm{d}z \, \left( \frac{\vartheta_2(z + \frac{x}{2} + \frac{y}{2}|i\tau_2)\vartheta_2(z - \frac{x}{2} - \frac{y}{2}|i\tau_2)}{\vartheta_2(z + \frac{x}{2} - \frac{y}{2}|i\tau_2)\vartheta_2(z - \frac{x}{2} + \frac{y}{2}|i\tau_2)} \right)^{-s}$$
$$\times \left( \frac{\vartheta_1(x|i\tau_2)\vartheta_1(y|i\tau_2)}{\vartheta_2(z + \frac{x}{2} - \frac{y}{2}|i\tau_2)\vartheta_2(z - \frac{x}{2} + \frac{y}{2}|i\tau_2)} \right)^{-t} \tag{6.3}$$

$$\sim \Gamma(1-t) \int \mathrm{d}\tau_2 \, \mathrm{d}x \, \mathrm{d}y \, \mathrm{d}z \, \mathrm{e}^{-sxy\partial_z^2 \log \vartheta_2(z|i\tau_2)} x^{-t} y^{-t} \left( \frac{2\pi\eta(i\tau_2)^3}{\vartheta_2(z|i\tau_2)} \right)^{-2t} \tag{6.4}$$

$$\sim \Gamma(1-t) \, s^{t-1} \log(s) \int \mathrm{d}\tau_2 \, \mathrm{d}z \, \left( \partial_z^2 \log \vartheta_2(z|i\tau_2) \right)^{t-1} \left( \frac{\vartheta_2(z|i\tau_2)}{2\pi\eta(i\tau_2)^3} \right)^{2t} . \tag{6.5}$$

The remaining integral does not simplify. We notice that for positive integer values of $t$ (which never corresponds to physical kinematics), this formula makes a lot of sense. Indeed, the formula has a simple poles there corresponding to the exchanged particles going on-shell. In fact, we would have expected double poles whose residues are equal to the mass-shifts of the amplitude. This is indeed the case, since the integral (6.2) also receives a contribution of order $s^{t-1}$ that has double poles at integer $t$. The remaining integral in (6.5) is the formula for the mass-shifts of the leading Regge trajectory as discussed in [14, 15, 21].

Eq. (6.5) is still rather complicated, since it still involves an integral over a sublocus of moduli space. As it stands, the integral (6.5) is not even well-defined if $t$ not a positive integer. The problem is that $\partial_z^2 \log \vartheta_2(z|i\tau_2)$ has zeros on the moduli space and thus the integrand has branch cuts. At these points in moduli space, our approximation breaks down since the exponent in (6.4) is no longer rapidly oscillating. This signals that we need to be more careful in doing this approximation.

**$s^t$ contribution.** It turns out that there is a more leading contribution to the amplitude, at least for real $s$. We saw already strong numerical evidence for this in [14]. It has to be extracted somewhat differently since this contribution is exponentially suppressed for imaginary values of $s$. The relevant idea for this was already discussed in [15]. First, we recall that the region $\tau_2 \to \infty$ includes the closed dilaton tadpole and thus we can only hope to get a sensible answer when we combine the diagram with the planar Möbius strip diagram. We can then deform the integration contour over $\tau_2$ as sketched in Figure 4. The vertical line at $\tau = i\tau_2$ is the integration contour for the annulus, while the vertical line at $\tau = \frac{1}{2} + i\tau_2$ is the Möbius strip contribution. At $\tau = i\infty$, the integrand has a pole and we take the principal value, which we isolated by closing the contour as in the picture. The two arcs near $\tau = 0$ and $\tau = \frac{1}{2}$ are the implementation of the $i\varepsilon$ contour in this context, see [14, 15, 22] for more details. These arcs are in particular necessary since we want to discuss the amplitude

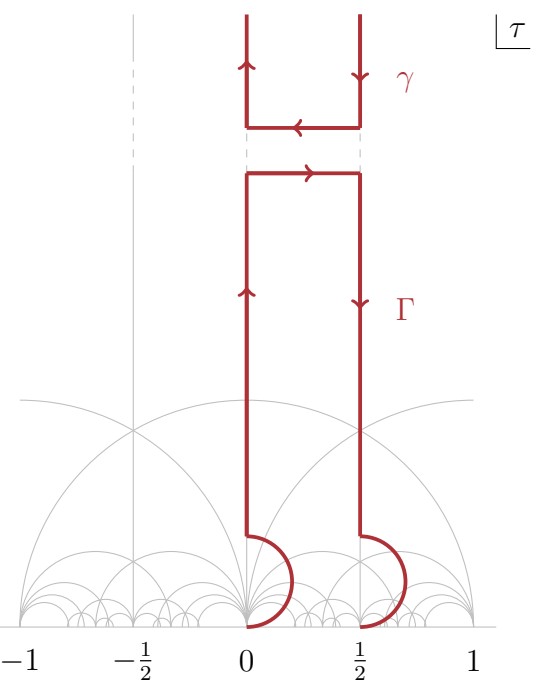

**Figure 4:** The integration contour for the planar amplitude.

for *real* $s$ and $t$. We now show that the contribution from $\gamma$ gives the desired leading contribution. The integral over $\Gamma$ is complicated, but we expect that it is of order $s^{t-1}\log(s)$ and is not enhanced. We do not know how to explicitly show this, but some numerical evidence for this was collected in [14, 15]. Indeed, we can compare the growth of the integral over $\Gamma$ with the growth of the integral over $\gamma$. The integral over $\gamma$ is simple to evaluate, see the discussion below and is asymptotically of order $s^t$ for large $s$. The imaginary part over the full contour can be evaluated explicitly via the Baikov representation explained in Section 3.2, where we sum over all diagrams with the planar color ordering (1234). We then subtract the integral over $\gamma$ to get access to the imaginary part of the contour over $\Gamma$. The numerical result is plotted in Figure 5 for three different values of $t$. One quite clearly sees that the rescaled integral asymptotically seems to oscillate around a constant, which confirms the expectation that it grows like $s^{t-1}\log(s)$. It would be nice to establish this analytically, but we will assume it from now on.

Thus, the leading contribution to the large $s$ asymptotics comes entirely from the $\gamma$ contour, which we will now evaluate. The contour $\gamma$ picks out the residue of the pole at $\tau = i\infty$, which in turn is given by an $\alpha'$-derivative of the tree-level amplitude [23]. This is a manifestation of the soft-dilaton theorem in string theory [24, 25]. Thus we obtain in a large $s$-limit with reinstated $\alpha'$

$$A^{\mathrm{P}} \sim -\frac{i\,\alpha'}{4\pi^2}\frac{\mathrm{d}}{\mathrm{d}\alpha'}\left((\alpha')^2\frac{\Gamma(-\alpha's)\Gamma(-\alpha't)}{\Gamma(1-\alpha's-\alpha't)}\right) \tag{6.6}$$

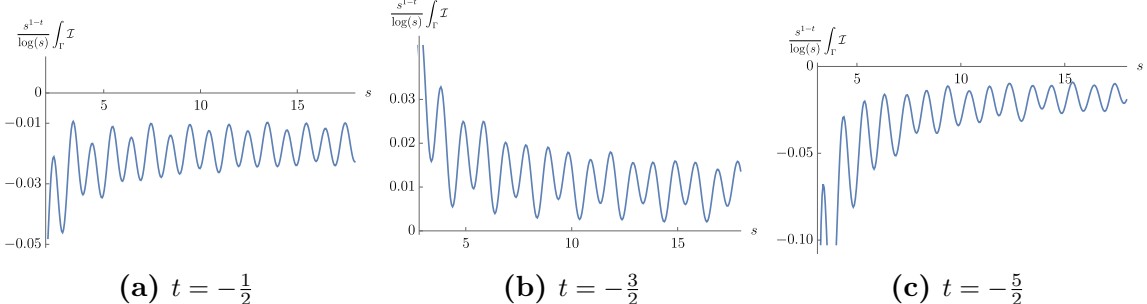

**(a)** $t = -\frac{1}{2}$         **(b)** $t = -\frac{3}{2}$         **(c)** $t = -\frac{5}{2}$

**Figure 5:** The integral over $\Gamma$. We divide by the expected growth behaviour $s^{t-1}\log(s)$ and observe that in all three cases $t = -\frac{1}{2}$, $t = -\frac{3}{2}$ and $t = -\frac{5}{2}$, the integral seems to approach an oscillatory function with constant mean.

$$\sim \frac{i\,\alpha'}{4\pi^2}\frac{\mathrm{d}}{\mathrm{d}\alpha'}\left((\alpha')^2(\alpha's)^{\alpha't-1}\Gamma(-\alpha't)\frac{\sin(\pi\alpha'(s+t))}{\sin(\pi\alpha's)}\right)\ . \tag{6.7}$$

The extra two factors of $\alpha'$ come from the polarization tensor $t_8$ which contains two powers of the Mandelstams which are accompanied by two powers of $\alpha'$. The biggest contribution comes when acting with the derivative on the trigonometric functions since it provides another power of $s$ from the inner derivative. We end up with

$$A^{\mathrm{p}} \sim \frac{i\,(\alpha')^2(\alpha's)^{\alpha't}}{4\,\Gamma(1+\alpha't)\,\sin(\pi\alpha's)^2} \tag{6.8}$$

This contribution is hence especially difficult to isolate: it is exponentially suppressed for imaginary values of $s$ and only visible for real $s$. It also does not exhibit poles in $t$ and cannot be naively related to the exchange of excitations on the leading Regge trajectory. We again notice that the contribution is purely imaginary.

**Other channels.** We can repeat this argument for the planar color ordering (1423). The situation is very similar. We get again an enhanced contribution from the cusp of the moduli space. The leading large-$s$ growth is of order

$$A^{\mathrm{p}} \sim \frac{i}{4\pi^2}\frac{\mathrm{d}}{\mathrm{d}\alpha'}\left((\alpha')^2(\alpha's)^{t-1}\frac{1}{\sin(\pi\alpha's)\Gamma(1+\alpha't)}\right) \tag{6.9}$$

$$\sim -\frac{i\,\alpha'(\alpha's)^t\,\cos(\pi\alpha's)}{4\,\Gamma(1+\alpha't)\,\sin(\pi\alpha's)^2}\ . \tag{6.10}$$

Similarly, this contribution does not have poles in $t$ and is exponentially suppressed for imaginary $s$.

Finally, let us note that the planar color ordering (1342) behaves quite differently. Indeed, while there is a cusp contribution to the amplitude as before, it does not feature trigonometric functions in the Regge limit and is thus only of order $s^{t-1}\log(s)$, which as discussed before also receives contributions from the rest of the integral.

This to be expected, since the imaginary part of the amplitude comes fully from the Möbius strip diagram (5.27) (there are no possible cuttings for the annulus with that color ordering). We showed that this cutting leads to the contribution (5.34) of order $s^{\frac{t}{2}-1}\log(s)^{-4}$ which dominates of $s^{t-1}\log(s)$ for $t<0$.

## 6.2 Higher-loop contributions

Let us again concentrate on the planar color ordering (1234). We will now extend the findings at one loop to higher loop orders. At any loop order $g$, there are $g$ holes in the open string diagram. Shrinking of the holes gives a closed string dilaton vertex operator inserted on the surface. For example at two loops, we add the four diagrams

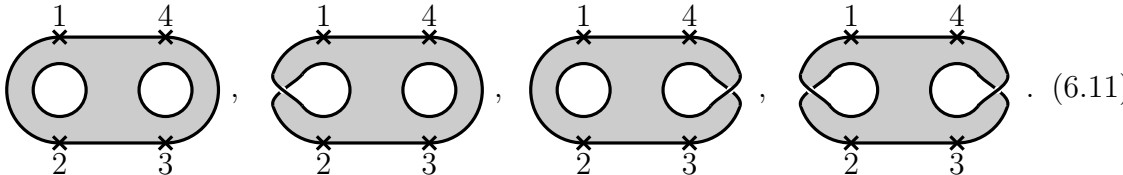

$$. \quad (6.11)$$

Adding the four diagrams cancels all the closed string dilaton tadpoles. In particular, all four integrals over the moduli space can be united into one integral which near the poles from the dilaton tadpoles are evaluated via a principal value prescription.

We can again analyze the Regge behaviour of this diagram using the same techniques as above. We split the integration contour into four parts analogous to the two parts at one-loop, see Figure 4. The simplest part is the contour in which we pick out the residue of the poles resulting from the dilaton tadpoles. Analogously to the situation at one loop, we assume those to be leading in the Regge limit (at fixed loop order). Picking out the residues from the dilaton tadpole gives

$$A^{\mathrm{p}}_{(2)} \sim \frac{1}{2}(-\pi i)^2 \; \underset{2 \qquad 3}{\overset{1 \qquad 4}{\boxed{\times \qquad \times}}} = \frac{1}{2}(-\pi i)^2 \left(\frac{\alpha'}{4\pi^3}\frac{\mathrm{d}}{\mathrm{d}\alpha'}\right)^2 \underset{2 \qquad 3}{\overset{1 \qquad 4}{\bigcirc}} . \quad (6.12)$$

Importantly, this formula has a factor of $\frac{1}{2}$ in front. It comes from the fact that the holes in the open string diagram are unlabeled and in particular exchanging the two holes leads to the same surface. On the other hand, we treat the closed string dilaton vertex operator insertions as labelled which instructs us to insert a factor of $\frac{1}{2!}=\frac{1}{2}$. The larges contribution appears again when the $\alpha'$ derivatives hit the trigonometric function $\frac{\sin(\pi\alpha'(s+t))}{\sin(\pi\alpha's)}$ in the Regge limit of the tree-level correlator.

Proceeding in the same way at $g$ loops gives similarly

$$A^{p}_{(g)} \sim \frac{1}{g!}(-\pi i)^g \left(\frac{\alpha'}{4\pi^3}\frac{\mathrm{d}}{\mathrm{d}\alpha'}\right)^g \underset{2 \qquad 3}{\overset{1 \qquad 4}{\bigcirc}} , \quad (6.13)$$

which scales like $s^{g-1+t}$ in the Regge limit at fixed loop order.

## 6.3 Genus resummation

Since we obtained the leading contribution to the Regge limit at a fixed loop order, we can attempt to perform the full sum over loops. To do it consistently, we have to be more careful about various factor of $i$ coming from the Wick rotation to Euclidean signature. These only lead to overall factors for a fixed loop order, but can be important when we sum over all genera.

Recall that the $S$-matrix is related to the $T$-matrix by $S = \mathbb{1} + iT$. The usual Feynman rules of QFT compute $T$. The counting of $i$'s is going to be the same as in the scalar $\phi^3$ theory, so let us use it as an example. The propagator in mostly-plus signature is $\frac{i}{-p^2-m^2+i\varepsilon}$ and the vertex is $-ig_{\rm s}$. Moreover, the loop integral runs over the Minkowski momenta. If we Wick rotate to Euclidean loop momenta, we pick up another factor of $i$ for every loop integration. In string theory, these factors of $i$ are missing since we are treating the worldsheet theory as a Euclidean CFT. In a diagram with $g$ loops, we have $2g-2+n$ vertices, $3g-3+n$ propagators, and $g$ loop integrals. Taking also the overall factor of $i$ into account, we should include an overall factor of

$$i\,(-i)^{2g-2+n}i^{3g-3+n}\,i^g = (-1)^g \tag{6.14}$$

for a $g$-loop diagram. The $\phi^3$ theory behaves exactly like string theory in this respect. Thus if we want to compute the string $S$-matrix, we should sum over the genus with an additional alternating sign.

We hence obtain for the string $S$-matrix

$$S \sim \sum_{g=0}^{\infty} \frac{(-1)^g}{g!} g_{\rm s}^{2g+2}(-\pi i)^g \left(\frac{\alpha'}{4\pi^3}\frac{\rm d}{{\rm d}\alpha'}\right)^g \quad \text{} \,, \tag{6.15}$$

where $g_{\rm s}$ is the open string coupling. For such a resummation to make sense, we need to require that every term in the sum is of the same order, which means that we are considering a double scaling limit of the form

$$s \to \infty \,, \qquad s\,g_{\rm s}^2 = {\rm const.} \tag{6.16}$$

Hence even though we are resumming the genus expansion, we still need to consider weak string coupling.

The resummation in this regime is now very simple:

$$S \sim g_{\rm s}^2 \exp\left(\frac{ig_{\rm s}^2\,\alpha'}{4\pi^2}\frac{\rm d}{{\rm d}\alpha'}\right) \quad \text{} \tag{6.17}$$

$$= g_{\rm s}^2 \quad \text{} \Big|_{\alpha'\to\alpha'\,{\rm e}^{\frac{ig_{\rm s}^2}{4\pi^2}}} \tag{6.18}$$

$$\sim -(\alpha')^2 g_{\mathrm{s}}^2 (\alpha' s)^{\alpha' t - 1} \Gamma(-\alpha' t) \frac{\sin(\pi \alpha' (s + \frac{i g_{\mathrm{s}}^2}{4\pi^2} s + t))}{\sin(\pi \alpha' (s + \frac{i g_{\mathrm{s}}^2}{4\pi^2} s))} \ . \tag{6.19}$$

We used that $\exp\left(x\alpha' \frac{\mathrm{d}}{\mathrm{d}\alpha'}\right)$ is the scaling generator in $\alpha'$-space and only kept the leading terms in the last line. As mentioned before, this leading term only deforms the trigonometric part of the amplitude.

Note that all poles moved into the lower half-plane of $s$, as expected after resummation. This provides an a posteriori justification for studying the Regge asymptotics at fixed-loop order with $s$ approaching the real axis from the upper half-plane.

We can in fact read off the mass spectrum of the leading Regge trajectory from the poles in the amplitude, which takes the form

$$M_n^2 = \frac{n}{\alpha'} \left(1 - \frac{i g_{\mathrm{s}}^2}{4\pi^2}\right) \ , \qquad n \in \mathbb{Z}_{\geqslant 0} \ . \tag{6.20}$$

Let us recall that this formula is valid for large $n$ with $n g_{\mathrm{s}}^2$ of order one, i.e. the corrections are small. Thus we conclude that at weak string coupling, the leading Regge trajectory slightly rotates into the complex plane and in particular the decay width is proportional to the mass level.

This computation is somewhat analogous to the usual resummation of 1PI diagrams in QFT, for which there is no clear analogue in string theory. Let us also mention that we have not carefully normalized the string coupling and thus the factor of $\frac{1}{4\pi^2}$ is essentially convention. In particular, we could have predicted this result from the one-loop mass-shift, which was computed in [14, 26] and then appeal to the usual 1PI resummation of field theory.

**Other channels and the closed string.** One may wonder why the analogous phenomenon does not occur in other channels or for the closed string. The reason is that these channels either do not have the infinite series of poles or this contribution is subleading. For example, in the closed string, the leading Regge behaviour at one-loop was

$$\frac{s^{3+t}}{\log(s)^4} \ , \tag{6.21}$$

which is always leading over the $\alpha'$ correction to the tree-level poles, which goes like $s^{3+2t}$. Thus it remains an interesting problem to determine the behaviour of the leading closed string Regge trajectory.

# 7 Conclusion

## 7.1 Discussion

While we believe that we achieved a significantly more complete understanding of the behaviour of string amplitudes, not all of our derivations are complete. We now

discuss the possible loopholes.

**Analytic continuation from the upper half plane.** One of the methods we employed was saddle point approximation. To have a well-defined starting point, this method assumes imaginary values for $s$ and $t$. Following [10], we noted that the results can be analytically continued to the whole upper and lower half-plane, respectively. Since we are mostly interested in the amplitude for real Mandelstam parameters, we had to make an educated guess for that limit. This limit is certainly not unique without further assumptions. For example, the function $\tan(\pi s)^2$ limits to

$$\tan(\pi s)^2 \longrightarrow 1 \tag{7.1}$$

when $|s|$ is taken radially to $\infty$ away from the real axis. Another function that has the same property is the constant function, so we cannot expect uniqueness on the real axis and thus the trigonometric factors appearing in (5.43) cannot be entirely fixed by our methods and have some degree of uncertainty.

**Subleading orders in the Regge limit.** The Regge limit discussed in this paper is the first term in an infinite series. One could in principle develop a systematic $\frac{1}{s}$ expansion of the amplitude, from both the Baikov representation and/or the saddle-point approximation. We expect that this expansion also has interesting number-theoretic properties, just as the low-energy expansion does.

## 7.2 Future directions

**High-energy fixed angle.** An arguably more interesting limit than the Regge limit was considered by Gross and Mende [8, 9], as well as Gross and Mañes for the open string [27]. In that limit, the amplitude at high energies for fixed scattering angle is considered (i.e. fixed value of $\frac{s}{t}$ for large $s$). In that case, one can also perform a saddle-point approximation of the moduli-space integral. However, such an evaluation has so far not been convincingly performed for the reasons mentioned in the introduction. The imaginary part of this limit can also be analyzed using the Baikov representation studied in this paper. It thus seems promising to us to revisit the high-energy limit of the one-loop amplitude using these new techniques.

**Other amplitudes.** We expect that there are straightforward extensions of the techniques analyzed in this paper to other situations. One can consider other topologies, such as the Klein bottle, higher point functions with multi-Regge kinematics, open string scattering amplitudes on D-branes or higher loop corrections. It is in particular interesting that type IIA and IIB amplitudes agree at one- and two-loops, but start to differ at higher loops. It is unclear to us what the imprint is on the Regge behaviour of those amplitudes.

**Sister trajectories.** Another curious direction for future research is that of sister Regge trajectories [28–30]. They correspond to Regge slopes which are halved (or more generally divided by an integer) compared to the standard trajectories and can be associated with "folded" worldsheet configurations [31]. It would be interesting to systematically explore their effects on one-loop amplitudes at higher multiplicity, perhaps along the lines of [32, 33].

**Dispersion relations.** The Regge asymptotics derived in this paper are a useful starting point to derive dispersion relations. In particular, the real part of the open-string scattering amplitude was analyzed in [15] using the Rademacher contour. Clearly, one could also try to use the growth behaviour of the amplitude together with the simpler imaginary part to obtain the real part directly using analyticity, thus bypassing the need for the Rademacher contour or other sophisticated tools.

**Linear Regge trajectories?** We have shown that the leading Regge trajectory remains a linear trajectory even at weak non-zero string coupling. With more work, it might be possible to demonstrate this also for subleading trajectories and/or to the next orders in the coupling. An old argument by Mandelstam suggests that linear trajectories are a consequence of narrow widths [34].

# Acknowledgements

We thank Nima Arkani-Hamed, Simon Caron-Huot, Aaron Hillman, Julio Parra-Martinez, Gabriele Veneziano and Sasha Zhiboedov for interesting discussions. P.B. would like to acknowledge the support provided by Fulbright Foundation (Award No. 2564/FNPDR/2020) and FAPESP Foundation through the grant 2023/03208-3. S.M. gratefully acknowledges funding provided by the Sivian Fund and the Roger Dashen Member Fund at the Institute for Advanced Study. This material is based upon work supported by the U.S. Department of Energy, Office of Science, Office of High Energy Physics under Award Number DE-SC0009988.

# A   The imaginary part of the one-loop closed string amplitude

In this appendix, we derive eq. (3.1). The derivation is very similar to what was explained in [14] and systematized in [15].

## A.1   contour for the imaginary part

Let us start by recalling that the closed string one-loop amplitude is given by

$$A = \int_{\mathcal{F}} \frac{\mathrm{d}^2\tau}{(\mathrm{Im}\,\tau)^5} \int_{\mathbb{T}^2} \prod_{j=1}^{3} \mathrm{d}^2 z_j \prod_{1 \leqslant i < j \leqslant 4} |\vartheta_1(z_{ij}|\tau)|^{-2s_{ij}}\, \mathrm{e}^{\frac{2\pi s_{ij}(\mathrm{Im}\,z_{ij})^2}{\mathrm{Im}\,\tau}} \ . \tag{A.1}$$

One can put $z_4$ to any fixed value by translation invariance of the torus. To get a well-defined integral, one also has to specify the behaviour of the integration contour more precisely near the degenerations of the surface. The most subtle region appears at $\operatorname{Im}\tau \to +\infty$. It is useful to parametrize $\tau$ and $z_j$ as follows:

$$\tau = \tau_x + \tau_y \ , \qquad z_j = x_j + \tau_y y_j \ . \tag{A.2}$$

On the naive contour, $\tau_x$, $x_j$ and $y_j$ are all purely real, while $\tau_y$ is purely imaginary. We then modify the contour of $\tau_y$ near $\tau_y \to +i\infty$ by letting it turn into the complex plane. As explained in [14], this is where the imaginary part comes from. The imaginary part is then isolated by taking the contour and subtracting the complex conjugate and dividing the result by 2. The imaginary part is thus given by

$$\operatorname{Im} A = \frac{1}{2} \int_{\longrightarrow} \frac{\mathrm{d}\tau_y}{\tau_y^2} \int_0^1 \mathrm{d}\tau_x \int_0^1 \prod_{j=1}^3 \mathrm{d}x_j\, \mathrm{d}y_j \prod_{1\leqslant j<i\leqslant 4} \vartheta_1(x_{ij}+\tau_y y_{ij}|\tau_x+\tau_y)^{-s_{ij}}$$
$$\times \vartheta_1(-x_{ij}+\tau_y y_{ij}|-\tau_x+\tau_y)^{-s_{ij}} \mathrm{e}^{-2\pi i s_{ij}\tau_y y_{ij}^2} \ . \tag{A.3}$$

## A.2  $q$-expansion

Let us denote $q = \mathrm{e}^{2\pi i \tau_y}$. Since we can localize the $\tau_y$ integral near the cusp $\tau_y = +i\infty$, it is useful to $q$-expand the integrand. With the help of the product representation of theta-functions, this leads to

$$\operatorname{Im} A = \frac{1}{2} \int_{\longrightarrow} \frac{\mathrm{d}\tau_y}{\tau_y^2} \int_0^1 \mathrm{d}\tau_x \int_0^1 \prod_{j=1}^3 \mathrm{d}x_j\, \mathrm{d}y_j \prod_{1\leqslant j<i\leqslant 4} q^{s_{ij}y_{ij}(1-y_{ij})}$$
$$\times \prod_{n\geqslant 1}\left(1-\mathrm{e}^{\pm 2\pi i(x_{ij}+\tau_x n)}q^{-y_{ij}+n}\right)^{-s_{ij}} \prod_{n\geqslant 0}\left(1-\mathrm{e}^{\pm 2\pi i(x_{ij}-\tau_x n)}q^{y_{ij}+n}\right)^{-s_{ij}} \ . \tag{A.4}$$

The $\pm$ sign in the exponent means that we are taking the product over both choices of signs.

The integral over $\tau_y$ only picks up regions in the $x_j$ and $y_j$ integration where the $q$-exponent is negative. This happens for positive $y_{31}$, $y_{41}$, $y_{32}$ and $y_{43}$. Thus, most terms in the $q$-expansion contribute positively to the exponent, except for the $n=0$ terms with $ij \in \{21, 43\}$. Let us separate them out and write

$$\operatorname{Im} A = \frac{1}{2} \int_{\longrightarrow} \frac{\mathrm{d}\tau_y}{\tau_y^2} \int_0^1 \mathrm{d}\tau_x \int_0^1 \prod_{j=1}^3 \mathrm{d}x_j\, \mathrm{d}y_j \left(1-\mathrm{e}^{\pm 2\pi i x_{21}}q^{y_{21}}\right)^{-s}\left(1-\mathrm{e}^{\pm 2\pi i x_{43}}q^{y_{43}}\right)^{-s}$$
$$\times \prod_{1\leqslant j<i\leqslant 4} q^{s_{ij}y_{ij}(1-y_{ij})} \prod_{n\geqslant 1}\left(1-\mathrm{e}^{\pm 2\pi i(x_{ij}+\tau_x n)}q^{-y_{ij}+n}\right)^{-s_{ij}}$$
$$\times \prod_{n\geqslant \delta_{ij,21}+\delta_{ij,43}}\left(1-\mathrm{e}^{\pm 2\pi i(x_{ij}-\tau_x n)}q^{y_{ij}+n}\right)^{-s_{ij}} \ . \tag{A.5}$$

We can thus now $q$-expand the terms in the second and third line. This leads to

$$
[q^{m_{\mathrm{L}}y_{21}+m_{\mathrm{D}}y_{32}+m_{\mathrm{R}}y_{43}+m_{\mathrm{U}}(1-y_{41})}]\prod_{n\geqslant 1}\left(1-\mathrm{e}^{2\pi i(x_{ij}+\tau_x n)}q^{-y_{ij}+n}\right)^{-s_{ij}}
$$

$$
\times\prod_{n\geqslant\delta_{ij,21}+\delta_{ij,43}}\left(1-\mathrm{e}^{-2\pi i(x_{ij}-\tau_x n)}q^{y_{ij}+n}\right)^{-s_{ij}}\tag{A.6}
$$

$$
=\mathrm{e}^{-2\pi i(m_{\mathrm{L}}x_{21}+m_{\mathrm{D}}x_{32}+m_{\mathrm{R}}x_{43}+m_{\mathrm{U}}(1-x_{41}))+2\pi i\tau_x m_{\mathrm{U}}}\left[q^{m_{\mathrm{L}}y_{21}+m_{\mathrm{D}}y_{32}+m_{\mathrm{R}}y_{43}+m_{\mathrm{U}}(1-y_{41})}\right]
$$

$$
\prod_{n\geqslant 1}\left(1-q^{-y_{ij}+n}\right)^{-s_{ij}}\prod_{n\geqslant\delta_{ij,12}+\delta_{ij,34}}\left(1-q^{y_{ij}+n}\right)^{-s_{ij}}\tag{A.7}
$$

$$
=\mathrm{e}^{-2\pi i(m_{\mathrm{L}}x_{21}+m_{\mathrm{D}}x_{32}+m_{\mathrm{R}}x_{43}+m_{\mathrm{U}}(1-x_{41}))+2\pi i\tau_x m_{\mathrm{U}}}\,Q_{m_{\mathrm{L}},m_{\mathrm{D}},m_{\mathrm{R}},m_{\mathrm{U}}}\;,\tag{A.8}
$$

where we made use of the definition (3.4). Proceeding analoguously with the expansion of the other sign choice in the infinite product leads to

$$
\operatorname{Im}A=\frac{1}{2}\int_{\longrightarrow}\frac{\mathrm{d}\tau_y}{\tau_y^2}\int_0^1\mathrm{d}\tau_x\int_0^1\prod_{j=1}^3\mathrm{d}x_j\,\mathrm{d}y_j\left(1-\mathrm{e}^{\pm 2\pi ix_{21}}q^{y_{21}}\right)^{-s}\left(1-\mathrm{e}^{\pm 2\pi ix_{43}}q^{y_{43}}\right)^{-s}
$$

$$
\times\prod_{1\leqslant j<i\leqslant 4}q^{s_{ij}y_{ij}(1-y_{ij})}\sum_{m_{\mathrm{L}},m_{\mathrm{D}},m_{\mathrm{R}},m_{\mathrm{U}}}\sum_{\tilde{m}_{\mathrm{L}},\tilde{m}_{\mathrm{D}},\tilde{m}_{\mathrm{R}},\tilde{m}_{\mathrm{U}}}Q_{m_{\mathrm{L}},m_{\mathrm{D}},m_{\mathrm{R}},m_{\mathrm{U}}}Q_{\tilde{m}_{\mathrm{L}},\tilde{m}_{\mathrm{D}},\tilde{m}_{\mathrm{R}},\tilde{m}_{\mathrm{U}}}
$$

$$
\times\mathrm{e}^{2\pi i((\tilde{m}_{\mathrm{L}}-m_{\mathrm{L}})x_{21}+(\tilde{m}_{\mathrm{D}}-m_{\mathrm{D}})x_{32}+(\tilde{m}_{\mathrm{R}}-m_{\mathrm{R}})x_{43}+(\tilde{m}_{\mathrm{U}}-m_{\mathrm{U}})(1-x_{41})-\tau_x(\tilde{m}_{\mathrm{U}}-m_{\mathrm{U}}))}
$$

$$
\times q^{(m_{\mathrm{L}}+\tilde{m}_{\mathrm{L}})y_{21}+(m_{\mathrm{D}}+\tilde{m}_{\mathrm{D}})y_{32}+(m_{\mathrm{R}}+\tilde{m}_{\mathrm{R}})y_{43}+(m_{\mathrm{U}}+\tilde{m}_{\mathrm{U}})(1-y_{41})}\;.\tag{A.9}
$$

We can integrate out $x_{32}$ and $\tau_x$, which imposes $\tilde{m}_{\mathrm{U}}=m_{\mathrm{U}}$ and $\tilde{m}_{\mathrm{D}}=m_{\mathrm{D}}$. Let us also rename $x_{21}\to x_{\mathrm{L}}$ and $x_{43}\to x_{\mathrm{R}}$. We also commute the sums and the integrals and set

$$
y_{21}=y_{\mathrm{L}}\;,\qquad y_{43}=y_{\mathrm{R}}\;,\qquad y_{31}=\frac{1}{s}(s-m_{\mathrm{D}}+uy_{\mathrm{R}}+t_{\mathrm{L}})\;,\tag{A.10}
$$

so that the new integration variables besides $\tau_y$ are $x_{\mathrm{L}}$, $x_{\mathrm{R}}$, $y_{\mathrm{L}}$, $y_{\mathrm{R}}$ and $t_{\mathrm{L}}$. We also integrate in

$$
1=\sqrt{\frac{-is\tau_y}{tu}}\int\mathrm{d}t_{\mathrm{R}}\;q^{\frac{1}{4stu}(st_{\mathrm{R}}-(s+2t)t_{\mathrm{L}}-2tuy_{\mathrm{R}}+(m_{\mathrm{D}}+m_{\mathrm{U}})t-st)^2}\;.\tag{A.11}
$$

This leads to

$$
\operatorname{Im}A=\frac{1}{2}\sum_{\substack{m_{\mathrm{L}},\tilde{m}_{\mathrm{L}},m_{\mathrm{D}},\\m_{\mathrm{R}},\tilde{m}_{\mathrm{R}},m_{\mathrm{U}}}}Q_{m_{\mathrm{L}},m_{\mathrm{D}},m_{\mathrm{R}},m_{\mathrm{U}}}Q_{\tilde{m}_{\mathrm{L}},m_{\mathrm{D}},\tilde{m}_{\mathrm{R}},m_{\mathrm{U}}}\int_{\longrightarrow}\frac{\mathrm{d}\tau_y}{\tau_y^2}\sqrt{\frac{-i\tau_y}{stu}}\int\mathrm{d}t_{\mathrm{L}}\,\mathrm{d}t_{\mathrm{R}}\;q^{-2P_{m_{\mathrm{D}},m_{\mathrm{U}}}}
$$

$$
\times\int_0^1\mathrm{d}x_{\mathrm{L}}\,\mathrm{d}y_{\mathrm{L}}\;\mathrm{e}^{2\pi ix_{\mathrm{L}}(m_{\mathrm{L}}-\tilde{m}_{\mathrm{L}})}q^{-y_{\mathrm{L}}(2t_{\mathrm{L}}-m_{\mathrm{L}}-\tilde{m}_{\mathrm{L}})}\left(1-\mathrm{e}^{\pm 2\pi ix_{\mathrm{L}}}q^{y_{\mathrm{L}}}\right)^{-s}
$$

$$
\times\int_0^1\mathrm{d}x_{\mathrm{R}}\,\mathrm{d}y_{\mathrm{R}}\;\mathrm{e}^{2\pi ix_{\mathrm{R}}(m_{\mathrm{R}}-\tilde{m}_{\mathrm{R}})}\,q^{-y_{\mathrm{R}}(2t_{\mathrm{R}}-m_{\mathrm{R}}-\tilde{m}_{\mathrm{R}})}\left(1-\mathrm{e}^{\pm 2\pi ix_{\mathrm{R}}}q^{y_{\mathrm{R}}}\right)^{-s}\;,\tag{A.12}
$$

where $P_{m_{\mathrm{D}},m_{\mathrm{U}}}$ was defined in (3.3). The integration domain can be changed to $0\leqslant x_{\mathrm{L,R}}\leqslant 1$ and $y_{\mathrm{L,R}}$ as well as $t_{\mathrm{L,R}}$ over the whole real line, since on the difference of

the two integration regions the exponent of $q$ is positive and thus doesn't contribute to the $\tau_y$-integral. We can now set

$$\zeta_{L,R} = e^{2\pi i(x_{L,R} + \tau_y y_{L,R})} \, , \tag{A.13}$$

so that the second line becomes

$$\frac{i}{4\pi^2 \tau_y} \int_{\mathbb{C}} d^2\zeta_L \, \zeta_L^{m_L - t_L - 1} \bar{\zeta}_L^{\tilde{m}_L - t_L - 1} |1 - \zeta_L|^{-2s} = \frac{i}{4\pi\tau_y} \frac{\gamma(m_L - t_L)\gamma(1-s)}{\gamma(1 + m_L - t_L - s)} \, , \tag{A.14}$$

where

$$\gamma(x) = \frac{\Gamma(x)}{\Gamma(1 - \tilde{x})} \, . \tag{A.15}$$

$\tilde{x}$ is the corresponding right-moving quantity to $x$. We can perform the integral over $\tau_y$ at this point, leading to the $P_{m_D,m_U}^{\frac{5}{2}}$ factor. We obtain

$$\begin{aligned}
\text{Im}\, A &= \frac{16\pi}{15\sqrt{stu}} \sum_{m_D,m_U} \int dt_L\, dt_R\, P_{m_D,m_U}^{\frac{5}{2}} \sum_{m_L,m_D,\tilde{m}_L,\tilde{m}_R} Q_{m_L,m_D,m_R,m_U} Q_{\tilde{m}_L,m_D,\tilde{m}_R,m_U} \\
&\quad \times (-t_L)_{m_L}(-t_R)_{m_R}(1-s-t_L+m_L)_{m_D+m_U-m_L}(1-s-t_R+m_R)_{m_D+m_U-m_R} \\
&\quad \times (-t_L)_{\tilde{m}_L}(-t_R)_{\tilde{m}_R}(1-s-t_L+\tilde{m}_L)_{m_D+m_U-\tilde{m}_L}(1-s-t_R+\tilde{m}_R)_{m_D+m_U-\tilde{m}_R} \\
&\quad \times \frac{\Gamma(-s)\Gamma(-t_L)\Gamma(-u_L)}{\Gamma(1+s)\Gamma(1+t_L)\Gamma(1+u_L)} \times \frac{\Gamma(-s)\Gamma(-t_R)\Gamma(-u_R)}{\Gamma(1+s)\Gamma(1+t_R)\Gamma(1+u_R)} \tag{A.16} \\
&= \frac{16\pi}{15\sqrt{stu}} \sum_{\substack{m_D,m_U \\ \sqrt{m_D}+\sqrt{m_U}\leqslant\sqrt{s}}} \int dt_L\, dt_R\, P_{m_D,m_U}^{\frac{5}{2}} Q_{m_D,m_U}^2 \\
&\quad \times \frac{\Gamma(-s)\Gamma(-t_L)\Gamma(-u_L)}{\Gamma(1+s)\Gamma(1+t_L)\Gamma(1+u_L)} \times \frac{\Gamma(-s)\Gamma(-t_R)\Gamma(-u_R)}{\Gamma(1+s)\Gamma(1+t_R)\Gamma(1+u_R)} \, . \tag{A.17}
\end{aligned}$$

In the last line, we used the definition of the polynomials $Q_{m_D,m_U}$ of eq. (3.5). We also used that the region defined by $P_{m_D,m_U} \geqslant 0$ is empty unless $\sqrt{m_D} + \sqrt{m_U} \leqslant \sqrt{s}$. We also made use of the definition (3.2). This recovers (3.1).

# B    Cutting rules of the open string

In this Appendix, we derive the imaginary parts of the different open string amplitudes. These are all simple applications of the formulas derived in [15] from the Rademacher expansion. We commit to $s$-channel cuts and do not display the color traces to make the expressions slightly less clumsy.

## B.1 Planar annulus

The imaginary part of the planar annulus is given by half of the Rademacher circle $\frac{0}{1}$, i.e.

$$\text{Im} \;\; \begin{matrix} 1 \\ \\ 2 \end{matrix} \Bigg) \Bigg( \hspace{-0.2cm} \Bigg) \Bigg( \begin{matrix} 4 \\ \\ 3 \end{matrix} = \frac{i}{2} A_{0,1}^{0,0,0,0} \tag{B.1}$$

$$= \frac{N\pi s^2}{60\sqrt{stu}} \sum_{\sqrt{m_{\mathrm{D}}}+\sqrt{m_{\mathrm{U}}} \leqslant \sqrt{s}} \int \mathrm{d}t_{\mathrm{L}}\,\mathrm{d}t_{\mathrm{R}}\; P_{m_{\mathrm{D}},m_{\mathrm{U}}}^{\frac{5}{2}} Q_{m_{\mathrm{D}},m_{\mathrm{U}}}$$

$$\times \frac{\Gamma(-s)\Gamma(-t_{\mathrm{L}})}{\Gamma(1+u_{\mathrm{L}})} \times \frac{\Gamma(-s)\Gamma(-t_{\mathrm{R}})}{\Gamma(1+u_{\mathrm{R}})} \;. \tag{B.2}$$

There is a second diagram that can be obtained by permuting punctures 3 and 4. This retains the $s$-channel, but changes the color ordering. We notice that

$$P_{m_{\mathrm{D}},m_{\mathrm{U}}}(s,u,t_{\mathrm{L}},t_{\mathrm{R}}) = P_{m_{\mathrm{D}},m_{\mathrm{U}}}(s,t,t_{\mathrm{L}},u_{\mathrm{R}}) \;, \tag{B.3a}$$

$$Q_{m_{\mathrm{D}},m_{\mathrm{U}}}(s,u,t_{\mathrm{L}},t_{\mathrm{R}}) = (-1)^{m_{\mathrm{D}}+m_{\mathrm{U}}} Q_{m_{\mathrm{D}},m_{\mathrm{U}}}(s,t,t_{\mathrm{L}},u_{\mathrm{R}}) \;. \tag{B.3b}$$

Thus we can bring the expression back into the original form by changing variables $u_{\mathrm{R}} \to t_{\mathrm{R}}$ in the integral. This leads to

$$\text{Im} \;\; \begin{matrix} 1 \\ \\ 2 \end{matrix} \Bigg) \Bigg( \hspace{-0.2cm} \Bigg) \!\! \times \!\! \begin{matrix} 4 \\ \\ 3 \end{matrix} = \frac{N\pi s^2}{60\sqrt{stu}} \sum_{\sqrt{m_{\mathrm{D}}}+\sqrt{m_{\mathrm{U}}} \leqslant \sqrt{s}} (-1)^{m_{\mathrm{D}}+m_{\mathrm{U}}} \int \mathrm{d}t_{\mathrm{L}}\,\mathrm{d}t_{\mathrm{R}}\; P_{m_{\mathrm{D}},m_{\mathrm{U}}}^{\frac{5}{2}} Q_{m_{\mathrm{D}},m_{\mathrm{U}}}$$

$$\times \frac{\Gamma(-s)\Gamma(-t_{\mathrm{L}})}{\Gamma(1+u_{\mathrm{L}})} \times \frac{\Gamma(-s)\Gamma(-u_{\mathrm{R}})}{\Gamma(1+t_{\mathrm{R}})} \;. \tag{B.4}$$

This shows that the $t$-channel diagram needs to get the sign as in eq. (3.8) in order for this to be compatible with cutting.

## B.2 Moebius strip

In the $s$-channel with planar color ordering, there are four Moebius strip contributions that can be read off from the Rademacher formula in [15]. They are given by $-\frac{i}{2}A_{1/2}^{1,0,0,0}$, $-\frac{i}{2}A_{1/2}^{0,1,0,0}$, $-\frac{i}{2}A_{1/2}^{0,0,1,0}$ and $-\frac{i}{2}A_{1/2}^{0,0,0,1}$ in the language of [15]. The relative minus sign comes from the opposite orientation of the contour as in the annulus case. The superscripts denote the 'winding number' of the color line on the left-, bottom-, right-

and top part of the gluing diagram. We find

$$\text{Im}\ \ = -\frac{i}{2}A_{1/2}^{0,0,1,0} \tag{B.5}$$

$$= -\frac{\pi s^2}{60\sqrt{stu}} \sum_{\sqrt{m_{\mathrm{D}}}+\sqrt{m_{\mathrm{U}}}\leqslant\sqrt{s}} \int \mathrm{d}t_{\mathrm{L}}\,\mathrm{d}t_{\mathrm{R}}\ P_{m_{\mathrm{D}},m_{\mathrm{U}}}^{\frac{5}{2}} Q_{m_{\mathrm{D}},m_{\mathrm{U}}}$$

$$\times \frac{\Gamma(-u_{\mathrm{L}})\Gamma(-t_{\mathrm{L}})\sin(\pi(s+t_{\mathrm{L}}))}{\Gamma(1+s)\sin(\pi s)} \times \frac{\Gamma(-u_{\mathrm{R}})\Gamma(-t_{\mathrm{R}})}{\Gamma(1+s)} \tag{B.6}$$

$$= \frac{\pi s^2}{60\sqrt{stu}} \sum_{\sqrt{m_{\mathrm{D}}}+\sqrt{m_{\mathrm{U}}}\leqslant\sqrt{s}} (-1)^{m_{\mathrm{D}}+m_{\mathrm{U}}} \int \mathrm{d}t_{\mathrm{L}}\,\mathrm{d}t_{\mathrm{R}}\ P_{m_{\mathrm{D}},m_{\mathrm{U}}}^{\frac{5}{2}} Q_{m_{\mathrm{D}},m_{\mathrm{U}}}$$

$$\times \frac{\Gamma(-s)\Gamma(-t_{\mathrm{L}})}{\Gamma(1+u_{\mathrm{L}})} \times \frac{\Gamma(-u_{\mathrm{R}})\Gamma(-t_{\mathrm{R}})}{\Gamma(1+s)} , \tag{B.7}$$

$$\text{Im}\ \ = -\frac{i}{2}A_{1/2}^{0,1,0,0} \tag{B.8}$$

$$= \frac{\pi s^2}{60\sqrt{stu}} \sum_{\sqrt{m_{\mathrm{D}}}+\sqrt{m_{\mathrm{U}}}\leqslant\sqrt{s}} (-1)^{m_{\mathrm{D}}+1} \int \mathrm{d}t_{\mathrm{L}}\,\mathrm{d}t_{\mathrm{R}}\ P_{m_{\mathrm{D}},m_{\mathrm{U}}}^{\frac{5}{2}} Q_{m_{\mathrm{D}},m_{\mathrm{U}}}$$

$$\times \frac{\Gamma(-s)\Gamma(-t_{\mathrm{L}})}{\Gamma(1+u_{\mathrm{L}})} \times \frac{\Gamma(-s)\Gamma(-t_{\mathrm{R}})}{\Gamma(1+u_{\mathrm{R}})} . \tag{B.9}$$

The other two diagrams follow from the obvious L $\leftrightarrow$ R or D $\leftrightarrow$ U replacement.

We can find the diagrams with different color ordering by permuting the operators 3 and 4 as in eq. (B.4). This replaces $t_{\mathrm{R}} \leftrightarrow u_{\mathrm{R}}$ and includes a factor $(-1)^{m_{\mathrm{D}}+m_{\mathrm{U}}}$ coming from (B.3b). Thus

$$\text{Im}\ \ = \frac{\pi s^2}{60\sqrt{stu}} \sum_{\sqrt{m_{\mathrm{D}}}+\sqrt{m_{\mathrm{U}}}\leqslant\sqrt{s}} \int \mathrm{d}t_{\mathrm{L}}\,\mathrm{d}t_{\mathrm{R}}\ P_{m_{\mathrm{D}},m_{\mathrm{U}}}^{\frac{5}{2}} Q_{m_{\mathrm{D}},m_{\mathrm{U}}}$$

$$\times \frac{\Gamma(-s)\Gamma(-t_{\mathrm{L}})}{\Gamma(1+u_{\mathrm{L}})} \times \frac{\Gamma(-t_{\mathrm{R}})\Gamma(-u_{\mathrm{R}})}{\Gamma(1+s)} , \tag{B.10}$$

$$\text{Im}\ \ = \frac{\pi s^2}{60\sqrt{stu}} \sum_{\sqrt{m_{\mathrm{D}}}+\sqrt{m_{\mathrm{U}}}\leqslant\sqrt{s}} (-1)^{m_{\mathrm{U}}+1} \int \mathrm{d}t_{\mathrm{L}}\,\mathrm{d}t_{\mathrm{R}}\ P_{m_{\mathrm{D}},m_{\mathrm{U}}}^{\frac{5}{2}} Q_{m_{\mathrm{D}},m_{\mathrm{U}}}$$

$$\times \frac{\Gamma(-s)\Gamma(-t_{\mathrm{L}})}{\Gamma(1+u_{\mathrm{L}})} \times \frac{\Gamma(-s)\Gamma(-u_{\mathrm{R}})}{\Gamma(1+t_{\mathrm{R}})} . \tag{B.11}$$

There is one more Moebius strip gluing diagram. It can be read off from the Rademacher expansion of the Moebius strip in the $u$-channel (exchanging labels 2 and 3 and hence $s \leftrightarrow u$). In this case there is only one term with winding numbers

$n_\mathrm{L} = 1$, $n_\mathrm{D} = -1$, $n_\mathrm{R} = 1$ and $n_\mathrm{U} = 0$. It takes the form

$$\mathrm{Im}\ \ \text{}\ \ = \frac{i}{2} A_{1/2}^{1,-1,1,0}\big|_{s\leftrightarrow u} \tag{B.12}$$

$$= -\frac{\pi s^2}{60\sqrt{stu}} \sum_{\sqrt{m_\mathrm{D}}+\sqrt{m_\mathrm{U}}\leqslant\sqrt{s}} (-1)^{m_\mathrm{D}+1} \int \mathrm{d}t_\mathrm{L}\, \mathrm{d}t_\mathrm{R}\ P_{m_\mathrm{D},m_\mathrm{U}}^{\frac{5}{2}} Q_{m_\mathrm{D},m_\mathrm{U}}$$

$$\times \frac{\Gamma(-t_\mathrm{L})\Gamma(-u_\mathrm{L})}{\Gamma(1+s)} \times \frac{\Gamma(-t_\mathrm{R})\Gamma(-u_\mathrm{R})}{\Gamma(1+s)} \ . \tag{B.13}$$

## B.3   Non-planar annulus

The $s$-channel non-planar annulus with color trace $\mathrm{tr}(t^{a_1}t^{a_2})\,\mathrm{tr}(t^{a_2}t^{a_3})$ admits two cuttings. They can be obtained from the $A_{0/1}^{n_1,n_2,n_3}$ with $n_{21}=0$ and $n_{43}=0$ or $n_{21}=0$ and $n_{43}=1$ term of [15, eq. (6.33)]. There are two more terms that are obtained by flipping the middle part of the cutting diagram as in eq. (3.7),

$$\mathrm{Im}\ \ \text{}\ \ = \frac{\pi s^2}{60\sqrt{stu}} \sum_{\sqrt{m_\mathrm{D}}+\sqrt{m_\mathrm{U}}\leqslant\sqrt{s}} (-1)^{m_\mathrm{D}+m_\mathrm{U}} \int \mathrm{d}t_\mathrm{L}\, \mathrm{d}t_\mathrm{R}\ P_{m_\mathrm{D},m_\mathrm{U}}^{\frac{5}{2}} Q_{m_\mathrm{D},m_\mathrm{U}}$$

$$\times \frac{\Gamma(-s)\Gamma(-t_\mathrm{L})}{\Gamma(1+u_\mathrm{L})} \times \frac{\Gamma(-s)\Gamma(-t_\mathrm{R})}{\Gamma(1+u_\mathrm{R})} \ , \tag{B.14}$$

$$\mathrm{Im}\ \ \text{}\ \ = \frac{\pi s^2}{60\sqrt{stu}} \sum_{\sqrt{m_\mathrm{D}}+\sqrt{m_\mathrm{U}}\leqslant\sqrt{s}} \int \mathrm{d}t_\mathrm{L}\, \mathrm{d}t_\mathrm{R}\ P_{m_\mathrm{D},m_\mathrm{U}}^{\frac{5}{2}} Q_{m_\mathrm{D},m_\mathrm{U}}$$

$$\times \frac{\Gamma(-s)\Gamma(-t_\mathrm{L})}{\Gamma(1+u_\mathrm{L})} \times \frac{\Gamma(-s)\Gamma(-u_\mathrm{R})}{\Gamma(1+t_\mathrm{R})} \ , \tag{B.15}$$

Finally, we have the following non-planar diagrams that can be extracted from the non-planar $u$-channel formula in [15].

$$\mathrm{Im}\ \ \text{}\ \ = \frac{\pi s^2}{60\sqrt{stu}} \sum_{\sqrt{m_\mathrm{D}}+\sqrt{m_\mathrm{U}}\leqslant\sqrt{s}} (-1)^{m_\mathrm{D}+m_\mathrm{U}} \int \mathrm{d}t_\mathrm{L}\, \mathrm{d}t_\mathrm{R}\ P_{m_\mathrm{D},m_\mathrm{U}}^{\frac{5}{2}} Q_{m_\mathrm{D},m_\mathrm{U}}$$

$$\times \frac{\Gamma(-t_\mathrm{L})\Gamma(-u_\mathrm{L})}{\Gamma(1+s)} \times \frac{\Gamma(-t_\mathrm{R})\Gamma(-u_\mathrm{R})}{\Gamma(1+s)} \ , \tag{B.16}$$

$$\mathrm{Im}\ \ \text{}\ \ = \frac{\pi s^2}{60\sqrt{stu}} \sum_{\sqrt{m_\mathrm{D}}+\sqrt{m_\mathrm{U}}\leqslant\sqrt{s}} \int \mathrm{d}t_\mathrm{L}\, \mathrm{d}t_\mathrm{R}\ P_{m_\mathrm{D},m_\mathrm{U}}^{\frac{5}{2}} Q_{m_\mathrm{D},m_\mathrm{U}}$$

$$\times \frac{\Gamma(-t_\mathrm{L})\Gamma(-u_\mathrm{L})}{\Gamma(1+s)} \times \frac{\Gamma(-t_\mathrm{R})\Gamma(-u_\mathrm{R})}{\Gamma(1+s)} \ . \tag{B.17}$$

The last diagram can be obtained by swapping punctures 3 and 4 which again gives the additional sign $(-1)^{m_\mathrm{D}+m_\mathrm{U}}$ from eq. (B.3b). This demonstrates the gluing rules explained in Section 3.2 for all channels directly from the worldsheet integral.

## C The large $s$ limit of $Q_{m_\mathrm{D},m_\mathrm{U}}$

In this Appendix, we demonstrate eq. (4.1). From the definition in eq. (3.4), we already see that only some terms can contribute in the large $s$ limit. These are the terms with $\ell = 1$, since terms with higher $\ell$ will contribute further down in the $q$-expansion without providing more powers of $s$. We can also replace $u$ by $-s$ in the exponent. The only contributing terms in the product are then

$$Q_{m_\mathrm{L},m_\mathrm{D},m_\mathrm{R},m_\mathrm{U}}(s,t) \sim [q_\mathrm{L}^{m_\mathrm{L}} q_\mathrm{D}^{m_\mathrm{D}} q_\mathrm{R}^{m_\mathrm{R}} q_\mathrm{U}^{m_\mathrm{U}}] \frac{(1 - q_\mathrm{L} q_\mathrm{D})^s (1 - q_\mathrm{R} q_\mathrm{D})^s (1 - q_\mathrm{L} q_\mathrm{U})^s (1 - q_\mathrm{R} q_\mathrm{U})^s}{(1 - q_\mathrm{D} q_\mathrm{R} q_\mathrm{U})^s (1 - q_\mathrm{D} q_\mathrm{L} q_\mathrm{U})^s}$$
$$\times (1 - q_\mathrm{D})^{-t} (1 - q_\mathrm{U})^{-t} . \tag{C.1}$$

We now notice that

$$[q^m] (1 + zq)^s = z^m \binom{s}{m} \sim \frac{z^m s^m}{m!} = [q^m] \mathrm{e}^{zqs} . \tag{C.2}$$

Since we are taking the large $s$ limit of every coefficient separately, this means that we may replace all factors involving $s$ exponents by the corresponding exponentials. Thus

$$Q_{m_\mathrm{L},m_\mathrm{D},m_\mathrm{R},m_\mathrm{U}}(s,t) \sim [q_\mathrm{L}^{m_\mathrm{L}} q_\mathrm{D}^{m_\mathrm{D}} q_\mathrm{R}^{m_\mathrm{R}} q_\mathrm{U}^{m_\mathrm{U}}] \mathrm{e}^{s(q_\mathrm{D} q_\mathrm{U} - q_\mathrm{D} - q_\mathrm{U})(q_\mathrm{R} + q_\mathrm{L})} (1 - q_\mathrm{D})^{-t} (1 - q_\mathrm{U})^{-t}$$
$$\tag{C.3}$$

$$= \frac{s^{m_\mathrm{L}+m_\mathrm{R}}}{m_\mathrm{L}! \, m_\mathrm{R}!} [q_\mathrm{D}^{m_\mathrm{D}} q_\mathrm{U}^{m_\mathrm{U}}] (q_\mathrm{D} q_\mathrm{U} - q_\mathrm{D} - q_\mathrm{U})^{m_\mathrm{L}+m_\mathrm{R}} (1 - q_\mathrm{D})^{-t} (1 - q_\mathrm{U})^{-t} .$$
$$\tag{C.4}$$

We can now perform the sum (3.5), which in the large $s$ limit becomes

$$Q_{m_\mathrm{D},m_\mathrm{U}}(s,t,t_\mathrm{L},t_\mathrm{R}) \sim \sum_{m_\mathrm{L},m_\mathrm{R}=0}^{m_\mathrm{D}+m_\mathrm{U}} Q_{m_\mathrm{L},m_\mathrm{D},m_\mathrm{R},m_\mathrm{U}}(s,t)(-t_\mathrm{L})_{m_\mathrm{L}}(-t_\mathrm{R})_{m_\mathrm{R}}(-s)^{2m_\mathrm{D}+2m_\mathrm{U}-m_\mathrm{R}-m_\mathrm{L}}$$
$$\tag{C.5}$$

$$\sim s^{2m_\mathrm{D}+2m_\mathrm{U}} [q_\mathrm{D}^{m_\mathrm{D}} q_\mathrm{U}^{m_\mathrm{U}}] \sum_{m_\mathrm{L},m_\mathrm{R}=0}^{m_\mathrm{D}+m_\mathrm{U}} \frac{(-t_\mathrm{L})_{m_\mathrm{L}}(-t_\mathrm{R})_{m_\mathrm{R}}(-1)^{m_\mathrm{L}+m_\mathrm{R}}}{m_\mathrm{L}! \, m_\mathrm{R}!}$$
$$\times (q_\mathrm{D} q_\mathrm{U} - q_\mathrm{D} - q_\mathrm{U})^{m_\mathrm{L}+m_\mathrm{R}} (1 - q_\mathrm{D})^{-t} (1 - q_\mathrm{U})^{-t} \tag{C.6}$$
$$= s^{2m_\mathrm{D}+2m_\mathrm{U}} [q_\mathrm{D}^{m_\mathrm{D}} q_\mathrm{U}^{m_\mathrm{U}}] (1 - q_\mathrm{D})^{-t+t_\mathrm{L}+t_\mathrm{R}} (1 - q_\mathrm{D})^{-t+t_\mathrm{L}+t_\mathrm{R}} . \tag{C.7}$$

Extracting the coefficient then leads to (4.1).

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
