# Peer review of "Regge Limit of One-Loop String Amplitudes"

_SciPost Physics_

## Round 1 · Referee Report · Anonymous (Referee 1) · 2024-6-3

Strengths

  1. Addressing an important problem
  2. Clearly written
  3. Sufficiently detailed explanation of the techniques employed

Weaknesses

  1. Misleading interpretation of some results
  2. Unclear conclusions to be drawn

Report

This paper deals with the important issue of the high energy behavior of perturbative string amplitudes with special attention given to the case of open-string collisions. It is a follow-up of previous works by two of the authors and is supposed to clarify and complete their previous (occasionally puzzling) claims.

The calculations are carried out in much detail and the techniques involved are clearly explained. In my opinion there is enough new material in the paper to justify its publication in this journal. On the other hand, I have strong reservations on the physical interpretation and presentation of some of the results. Given the goals of the paper, it looks necessary for the authors to answer such concerns. Hopefully, by doing so, the paper will be improved.

My main criticism concerns the argument given after eq. (1.3) as an explanation for the O(s) enhancement of certain open-string loop diagrams with respect to QFT expectations.
The authors claim that this enhancement originates from loop contributions appearing as derivatives of (1.3) with respect to \alpha’. It is claimed that such derivatives are well-behaved except if they act on the ratio of sin factors present in (1.3).
However, it easy to show that such a pathological consequence of the \alpha’ derivative only occurs on the real-s axis and, of course, only at t different from zero (hence producing a surprising dip in the imaginary part at t=0?).

On the other hand, it is well known that the asymptotic behavior of tree-level string amplitudes is ill-defined on the real axis whenever one encounters poles on it all the way to infinity.
The correct procedure is to take the limit after a slight rotation in the complex-s plane. It is immediate to prove that, by so doing, the ratio of the two sinus terms yields just the correct phase of Regge behavior, the one spelled out by the authors themselves in Sect. 2.

Once the \alpha’ derivatives are applied to such a Regge-behaved amplitude it no longer generates an O(s) enhancement but only a logarithmic one (coming from the non-trigonometric part in (1.3)). The authors appear to be aware of this delicate point when they mention a “possible loophole” at page 36 (“Analytic continuation from the upper half plane”). In my opinion this is not just a possible loophole but rather the loophole in their claim.

Using the above procedure, the results presented in Fig. 1 will still hold as far as the first and last line (corresponding to no flavor exchange in the t-channel) and the 3rd, 5th and 6th line (corresponding to the exchange of two mesons in the t-channel with the typical two-Reggeon-cut behavior) are concerned. Instead, the 2nd and 4th lines get modified in a way that reconciles them with more conventional expectations. It also reconciles the corrected (in the way explained above) perturbative results for the planar loops with the expansion of their resummed expression discussed in Sect. 6. justifying the statement made after eq. (1.4).

In essence, the resummation of the planar loops justifies, a posteriori, the rotation in the complex plane needed to recover the conventional Regge limit at tree-level: after the resummation one can stay instead on the real axis since the poles have been shifted to complex values of s (presumably on the second Riemann sheet).

In my opinion the paper, as presently written, is confusing, if not misleading. Staying away from the real axis from the beginning will bring about a much clearer and uncontroversial picture of their otherwise interesting results.

Requested changes

A major revision of the presentation and interpretation of the results is needed

Recommendation

Ask for major revision

---

## Editorial Decision

resubmitted